# Development of an online-coupled MARGA upgrade for the two hours interval quantification of low-molecular weight organic acids in the gas and particle-phase

Bastian Stieger[1], Gerald Spindler[1], Dominik van Pinxteren[1], Achim Grüner[1], Markus Wallasch[2], Hartmut Herrmann[1,*]

[1]Atmospheric Chemistry Department (ACD), Leibniz Institute for Tropospheric Research (TROPOS), Permoserstraße 15, 04318 Leipzig, Germany
[2]German Federal Environment Agency, Wörlitzer Platz 1, 06844 Dessau-Roßlau, Germany

*Correspondence to*: Hartmut Herrmann (herrmann@tropos.de)

**Abstract.** A method is presented to quantify the low-molecular weight organic acids formic, acetic, propionic, butyric, pyruvic, glycolic, oxalic, malonic, succinic, malic, glutaric, and methanesulfonic acid in the atmospheric gas and particle-phase, based on a combination of the Monitor for AeRosols and Gases in ambient Air (MARGA) and an additional ion chromatography (Compact-IC) instrument. Therefore, every second hourly integrated MARGA gas and particle-sample were collected and analyzed by the Compact-IC resulting in 12 values per day for each phase. A proper separation of the organic target acids was initially tackled by a laboratory IC optimization study, testing different separation columns, eluent compositions and eluent flow rates both for isocratic and gradient elution. Satisfactory resolution of all compounds was achieved using a gradient system with two coupled anion-exchange separation columns. Online pre-concentration with an enrichment factor of approximately 400 was achieved by solid-phase extraction consisting of a methacrylate polymer based sorbent with quaternary ammonium groups. The limits of detection of the method range between 7.1 ng m$^{-3}$ for methanesulfonate and 150.3 ng m$^{-3}$ for pyruvate. Precisions are below 1.0 %, except for glycolate (2.9 %) and succinate (1.0 %). Comparisons of inorganic anions measured at the TROPOS research site in Melpitz, Germany, by the original MARGA and the additional Compact-IC are in agreement with each other ($R^2$ = 0.95 - 0.99). Organic acid concentrations from May 2017 as an example period are presented. Monocarboxylic acids were dominant in the gas-phase with mean concentrations of 553 ng m$^{-3}$ for acetic acid, followed by formic (286 ng m$^{-3}$), pyruvic acid (182 ng m$^{-3}$), propionic (179 ng m$^{-3}$), butyric (98 ng m$^{-3}$) and glycolic (71 ng m$^{-3}$). Particulate glycolate, oxalate and methanesulfonate were quantified with mean concentrations of 63 ng m$^{-3}$, 74 ng m$^{-3}$ and 35 ng m$^{-3}$, respectively. Elevated concentrations in the late afternoon of gas-phase formic acid and particulate oxalate indicate photochemical formation as a source.

## 1 Introduction

Low-molecular weight organic acids have been measured in the gas (Lee et al., 2009;Bao et al., 2012) and particle-phase (Boreddy et al., 2017;Miyazaki et al., 2014;van Pinxteren et al., 2014) as well as in precipitation and cloud water (Sun et al., 2016;van Pinxteren et al., 2005). Next to known primary anthropogenic (Bock et al., 2017;Kawamura and Kaplan, 1987;Legrand et al., 2007) and biogenic sources (Falkovich et al., 2005;Stavrakou et al., 2012), organic acids are formed as secondary products by atmospheric oxidation processes (Lim et al., 2005;Tilgner and Herrmann, 2010;Hoffmann et al., 2016). However, there are still unknown sources of these short-chained compounds (Millet et al., 2015;Stavrakou et al., 2012). Because of their hygroscopicity (Kawamura and Bikkina, 2016), the organic acids contribute to the acidity of precipitation, dew, fog and clouds (Lee et al., 2009;van Pinxteren et al., 2016). Atmospheric transport processes also lead to dry and wet deposition in remote areas, where they can have an influence on the sensitive ecosystem (Friedman et al., 2017;Himanen et al., 2012;Sabbioni et al., 2003).

Owing to the low concentrations and the high diversity of organic acids compared to inorganic compounds, a highly resolved and near-real-time quantification of organic acids is challenging. Studies on organic compounds in particulate matter (PM) were performed with filter measurements followed by off-line analysis with ion chromatography (IC) (Röhrl and Lammel, 2002;Granby et al., 1997;Legrand et al., 2007), gas chromatography coupled with mass spectroscopy (GC/MS) (Mochizuki et al., 2018;Miyazaki et al., 2014;Kawamura et al., 2012;Hu et al., 2018) or flame ionization detector (GC-FID) (Deshmukh et al., 2018), capillary electrophoresis (CE) (Müller et al., 2005;van Pinxteren et al., 2014;van Pinxteren et al., 2009) or Raman spectroscopy (Kuo et al., 2011).

Gas-phase compounds were sampled for a few hours and analyzed off-line with coated filters and GC/MS (Limbeck et al., 2005), denuder and GC/MS (Bao et al., 2012), denuder and IC (Dawson et al., 1980), as well as a mist chamber and IC (Preunkert et al., 2007;Schultz Tokos et al., 1992), respectively.

Due to the long sampling time of filter and wet sampling techniques followed by laboratory analyses, these methods did not allow for a near-real-time quantification and the laboratory effort is huge. Recently, Stieger et al. (2018) showed that off-line filter analysis involves the risk of possible evaporation artifacts of volatile particulate compounds from the filter or the adsorption of gaseous compounds. Additionally, Boring et al. (2002) mentioned the difficulty of sampling very small particles by impaction techniques.

Over the last few years, new instruments have allowed online measurements with increased time resolution. Zander et al. (2010) and Pommier et al. (2016) quantified the vertical column of gaseous formic acid with ground-based Fourier transform infrared spectroscopy (FTIR). However, the focus of the present work is on the ground-based detection of the carboxylic acids because of possible influences on the lower troposphere.

Gas-phase concentrations on the ground were monitored with a Chemical Ionisation Mass Spectrometer (CIMS) (Veres et al., 2011;Liu et al., 2012a;Crisp et al., 2014;Mungall et al., 2018). This instrument also enabled airborne measurements of formic

acid (Jones et al., 2014). Recently, Nah et al. (2018b) assessed the use of sulfur hexafluoride ($SF_6^-$) anions as CIMS reagent ions as it is more sensitive for the detection of oxalic, propionic and glycolic acid.

As all organic acids are ionic, an application of the IC for the analysis is obvious. Boring et al. (2002) first described an instrument based on an IC system. The separation of the gas and particle-phase was performed by the application of a parallel plate denuder and a particle collection system consisting of glass fiber filters. The filters were washed online with deionized water and the dissolved anions from the gas and particle-phase, including formic, acetic and oxalic acid, were analyzed. The resulting time resolution from their example measurement period was approximately 30 minutes. A disadvantage in this study was the necessary exchange of the inserted glass fiber filters every 12 hours. Fisseha et al. (2006) published results of formic, acetic, propionic and oxalic acid in Zurich, Switzerland, for three months in different seasons. These authors used a flattened denuder and an aerosol chamber under supersaturated conditions to quantify formate, acetate, propionate and oxalate in the gas and particle-phase. The detection of other dicarboxylic acids (DCA) was not possible due to co-elution with the carbonate peak and atmospheric concentrations of other monocarboxylic acids (MCA) were mostly below the detection limit of the method. Lee et al. (2009) and Ku et al. (2010) sampled only gaseous compounds with a parallel plate denuder. While the first group analyzed C1-C3 MCA within an hourly time resolution, the second group concentrated on the quantification of acetic acid every ten minutes. Recently, Zhou et al. (2015) observed gaseous and particulate oxalate in their MARGA (Monitor for AeRosols and Gases in ambient Air) measurements in Hong Kong for one year. In this case, a pre-concentration column was installed instead of the injection loop, but the analysis of more carboxylic acids (CA) was limited by the short separation column and, thus, separation efficiency.

Recently, Nah et al. (2018a) presented measurements of low-molecular weight organic acids within the gas and particle-phase with use of a CIMS and a Particle-Into-Liquid-Sampler (PILS) coupled with a capillary high-pressure ion chromatography (HPIC), respectively. They received hourly concentrations of these compounds in a rural southeastern United States site for one month and were able to investigate the gas-particle partitioning.

Ullah et al. (2006) developed an on-line instrument to measure ionic species within the gas and particle-phase. For the separation, they used a membrane denuder to collect the water-soluble gases and a hydrophilic filter sampled the particles. In their ion chromatography analysis, it was possible to quantify formic and acetic acid every 40 minutes.

However, to the author's knowledge, online instruments properly quantifying a variety of low-molecular weight organic acids (formic, acetic, propionic, butyric, glycolic, pyruvic, oxalic, malonic, succinic, malic, glutaric, and methanesulfonic acid) within the gas and particle-phase in a high time resolution do not exist yet.

The present study describes the instrumental development of an online-coupled pre-concentration and ion chromatographic (IC) separation system to determine organic acids in the gas and particle-phase as an extension of the MARGA. The MARGA has been reported a reliable field instrument for long-time measurements in Melpitz and other sites (Stieger et al., 2018 and references therein) and its upgrade with an additional IC separation allows for the analysis of all target compounds with low risk of interferences from other species.

The developed setup was employed from November 2016 until October 2017 at the TROPOS research site in Melpitz. As a demonstration of a successful field application, the first tropospheric measurements will be presented. Data interpretation of the one year measurement campaign with a focus on the phase distribution and the investigation of primary and secondary sources will be published elsewhere.

## 2 Instrumentation and materials

### 2.1 MARGA

Water-soluble chloride ($Cl^-$), nitrate ($NO_3^-$), sulphate ($SO_4^{2-}$), ammonium ($NH_4^+$), sodium ($Na^+$), potassium ($K^+$), magnesium ($Mg^{2+}$) and calcium ($Ca^{2+}$) in particles smaller than 10 µm ($PM_{10}$) as well as the trace gases hydrochloric acid (HCl), nitrous acid (HONO), nitric acid ($HNO_3$), sulfur dioxide ($SO_2$) and ammonia ($NH_3$) are quantified hourly by the commercial MARGA 1S ADI 2080 (Metrohm-Applikon, The Netherlands) (Chen et al., 2017). Its technical principles and long-term operation at the TROPOS research site in Melpitz (Spindler et al., 2013;Spindler et al., 2004) were recently described in Stieger et al. (2018). Briefly, the separation of the gas and particle-phase is performed through the usage of a Wet Rotating Denuder (WRD) and a Steam-Jet Aerosol Collector (SJAC), respectively. Both the WRD and SJAC are continuously filled with an absorption solution. For one hour, syringe pumps sample 25 ml of the liquid solutions of the WRD and SJAC. Within the next hour, approximately 7 ml of each solution are transported to two ion chromatography systems to quantify the inorganic anions and cations in both phases. The remaining sample solution is directly discarded. A continuous calibration with an internal standard (lithium bromide) is applied.

### 2.2 Additional IC system

In addition to the two IC systems integrated into the MARGA, an additional one (Compact-IC Flex 930, Metrohm, Switzerland; further named as Compact-IC) together with an autosampler (Robotic Sample Processor XL, Metrohm, Switzerland) is used for the determination of organic acids. The setup of the complete system is shown in Fig. 1. Therein, the different components that will be explained in the following are tagged. Comparable IC systems, for example from Thermo Scientific, were considered as possible alternatives. However, the liquid handling via the autosampler, especially the liquid flows from the MARGA to the necessary autosampler and the capacity of the autosampler, limited the use of other IC systems.

An autosampler with two working stations (a) and (b) has a sample plate with 120 slots for 12.5 ml vials with perforated plugs (polypropylene; Metrohm, Switzerland). The slots are arranged in two circles. Additionally, one working station is equipped with inner and outer sample needles (a) so that the WRD and SJAC solutions can be pumped into the respective vial simultaneously. After storage, the filled vials go to the second working station consisting of a swing head with a further sample needle (b). To avoid contamination, this sample needle is cleaned in a washing station with ultrapure water after each suction.

A commercial syringe pump (DOSINO 800, Metrohm, Switzerland; c) transports 10 ml of one sample from the autosampler via a 6-way injection valve within the Compact-IC (d) to a sample loop (e) with a velocity of 2 ml min$^{-1}$. A graphical

explanation of the different modes of the 6-way-valve is in the supplement (Fig. S1). Afterwards, the injection valve switches to the fill mode and the DOSINO 800 transfers the complete sample volume into a pre-concentration column (Metrosep A PCC 2 VHC, Metrohm, Switzerland) consisting of a spherical methacrylate polymer with quaternary ammonium groups in the Compact-IC. In the injection mode, the degassed eluent consisting of 7 mM sodium carbonate ($Na_2CO_3$) / 0.75 mM sodium

hydroxide (NaOH) desorbs the trapped ionic compounds from the pre-concentration column with a flow of 0.8 ml min$^{-1}$ while the sample path is rinsed with ultrapure water. The anion-exchange separation column (Metrosep A Supp 16 250 mm, Metrohm, Switzerland) is stored in a column oven. Before the ionic signals are measured with a conductivity detector, the background conductivity of the anion eluent is chemically suppressed using 100 mM sulfuric acid ($H_2SO_4$) and 20 mM oxalic acid.

For gradient applications, a second DOSINO 800 (f) was combined with the Compact-IC. With an identical flow rate, a defined amount of a higher concentrated eluent was added to the eluent flow in front of the eluent degasser. A trap column (Metrosep A Trap 1 100/4.0) cleans and ensures a complete mixing of both eluent solutions before the eluent is injected into the pre-concentration column. For the combination of the MARGA and Compact-IC, an external 6-way-valve (Metrohm, Switzerland) is required (g). The complete setup and the time program for the gradient system is controlled by the Metrohm MagIC Net

software (Metrohm, Switzerland).

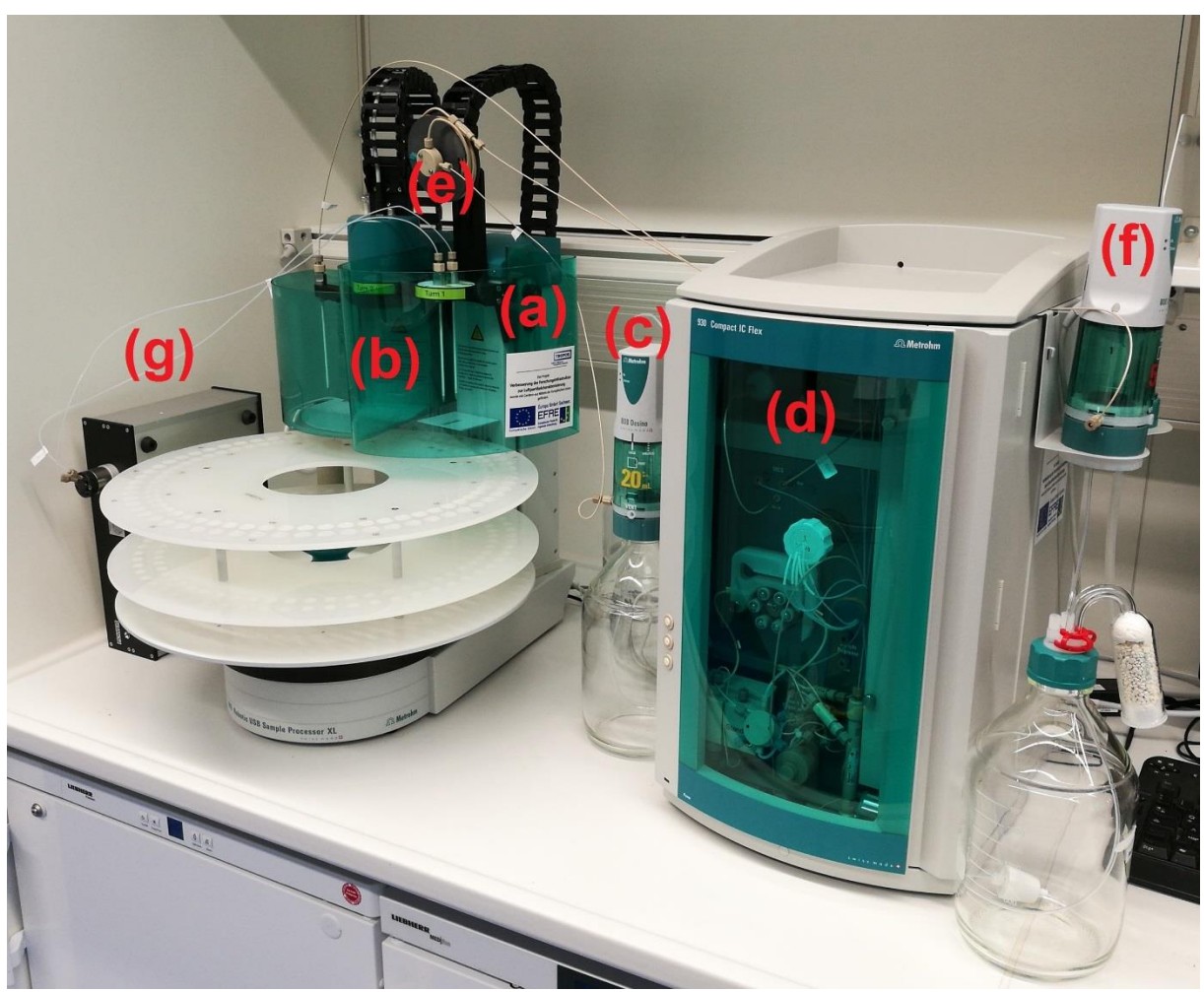

**Figure 1.** Setup of the IC-system with (a) the first working station and (b) the second working station of the autosampler, (c) the DOSINO 800 for the sample transportation, (d) the Compact-IC, (e) the 10 ml sample loop, (f) the DOSINO 800 for the gradient system and (g) an external 6-way-valve for the combination of MARGA and the IC-system.

The Compact-IC is manually calibrated with three standard solutions twice a week, when the vials of the autosampler are replaced. The standard solutions are prepared in 50 ml flasks, stored in the refrigerator and renewed every two weeks. The concentrations of each inorganic and organic ion in the standard are given in Table 1.

**Table 1.** Aqueous standard solution concentrations used for the calibration of the Compact-IC.

| Ions | Standard 1 / µg l$^{-1}$ | Standard 2 / µg l$^{-1}$ | Standard 3 / µg l$^{-1}$ |
|---|---|---|---|
| F$^-$ | 1 | 10 | 30 |
| Cl$^-$ | 5 | 50 | 150 |
| NO$_2^-$ | 5 | 50 | 150 |
| Br$^-$ | 1 | 10 | 30 |
| NO$_3^-$ | 10 | 100 | 300 |
| SO$_4^{2-}$ | 5 | 50 | 150 |
| Formate | 5 | 50 | 150 |
| Acetate | 5 | 50 | 150 |
| Propionate | 3 | 10 | 25 |
| Butyrate | 3 | 10 | 25 |
| Pyruvate | 3 | 10 | 25 |
| Glycolate | 3 | 10 | 25 |
| Oxalate | 3 | 10 | 25 |
| Malonate | 3 | 10 | 25 |
| Succinate | 3 | 10 | 25 |
| Malate | 3 | 10 | 25 |
| Methanesulfonate | 3 | 10 | 25 |

## 2.3 Materials

Hydrogen peroxide (H$_2$O$_2$, 30 %, Fluka) is used for the preparation of the MARGA absorption and cleaning solution. The MARGA anion and cation eluents are aqueous solutions of sodium carbonate monohydrate (99.5 %, Sigma Aldrich) and sodium bicarbonate (99.7 %, Sigma Aldrich) as well as 2 M nitric acid solution (Fluka), respectively. The MARGA internal standard and the suppressor regenerant solution are prepared with lithium bromide (99 %, Fluka) and with phosphoric acid (85 %, Fluka), respectively. For the Compact-IC eluent, sodium carbonate (Na$_2$CO$_3$) (99.5 %, Sigma Aldrich) and sodium hydroxide (NaOH) solution (50-52 %, Fluka) is dissolved. Sulfuric acid (98 %, ChemSolute) and oxalic acid (99 %, Sigma Aldrich) are mixed for the suppressor regenerant solution. The following chemicals are used for peak identification and calibration: fluoride (F$^-$), chloride, nitrite (NO$_2^-$), bromide (Br$^-$), nitrate, sulphate, formate, acetate, oxalate, methanesulfonate standards for IC (all 1000 mg l$^{-1}$, Fluka), propionic acid (99.5 %, Fluka), butyric acid (99 %, Aldrich), pyruvic acid (98 %, Aldrich), glycolic acid (99 %, Fluka), malonic acid (99 %, Fluka), succinic acid (99.5 %, Fluka), malic acid (99 %, Fluka) and glutaric acid (98 %, Fluka). All chemicals are dissolved in ultrapure water (18.2 MΩcm).

## 3 Results and discussion

### 3.1 Development of the IC separation

The IC separation was developed in laboratory studies to ensure the best separation efficiency of the target compounds formate, acetate, glycolate, pyruvate, oxalate, malonate, succinate, malate, and glutarate. The further organic anions propionate, butyrate and methanesulfonate were later identified in the first field applications and then included into the standard solution. Due to their expected low concentrations, it was considered important to pre-concentrate the ions online within the aqueous MARGA sample streams from both the WRD and SJAC. Therefore, a pre-concentration column was applied from the beginning of the optimization studies, as described above. An enrichment factor of 400 was achieved by the comparison of the peak areas of standard solutions applying a 20 µl injection loop and the pre-concentration column.

First analyses were performed with an isocratic system and the separation column Metrosep A Supp 16 250 mm with an eluent of 7 mM $Na_2CO_3$ and 0.75 mM NaOH. The resulting chromatogram is shown in Fig. 2, based on aqueous standards with concentrations of 10 µg l$^{-1}$ (Cl$^-$, NO$_3^-$, SO$_4^{2-}$), 5 µg l$^{-1}$ (NO$_2^-$) and 1 µg l$^{-1}$ (F$^-$, Br$^-$, all organic acids). The standards were loaded with a volume of 10 ml on the pre-concentration column and then desorbed into the separation column as described above. Regarding the MARGA system, these liquid concentrations would correspond to the mass concentrations of 250 ng m$^{-3}$, 125 ng m$^{-3}$ and 25 ng m$^{-3}$, respectively. The chosen organic acid concentrations were in agreement with impactor measurements sampled in Melpitz (van Pinxteren et al., 2014). However, there was no baseline-separation of acetate and Cl$^-$ and the concentrations of the inorganic compounds can exceed 10 µg m$^{-3}$, resulting in wider peaks and co-elution. This behaviour was observed for SO$_4^{2-}$ and oxalate as well as for the first peaks between F$^-$ and Cl$^-$ (Fig. S2).

Since, at this stage, a satisfying separation was not achieved, other columns were additionally tested within the isocratic setup. An anion-exchange column named Shodex IC SI-50 4E (Showa Denko Europe GmbH, Germany) was included with an eluent of 3.2 mM $Na_2CO_3$ and 1 mM $NaHCO_3$ and an operating temperature of 30 °C. The resulting chromatogram is shown in Fig. 3a. As can be seen, almost all CAs co-eluted in two distinct peaks (peaks A and B in Fig. 3a). The MCAs formate, acetate, glycolate and pyruvate could not be separated as well as SO$_4^{2-}$ with the DCAs malonate, succinate, malate and glutarate. As a result, this column was discarded. A Metrosep A Supp 7 250 mm (Metrohm, Switzerland) was tested with an eluent consisting of 3.2 mM $Na_2CO_3$ at 45 °C. The target MCAs in the chromatogram in Fig. 3b eluted close together between seven and nine minutes and a baseline-separation was not achieved. Regarding the low standard concentrations, the separation can be expected to worsen for high concentrations with this anion-exchange column. Higher ion concentrations would broaden the single peaks, which leads to co-elution. The advantage of the last two columns was the excellent separation of oxalate. However, as the aim of this work was the detection of all the target organic acids and the initial Metrosep A Supp 16 provided a good separation of all target compounds with no co-elution, this column was chosen for further improvements of the separation.

Possible improvements were investigated by changing the eluent flow and the eluent composition, which are summarized in Table 2. The flow was increased to 0.9 and 1.0 ml min$^{-1}$ and decreased to 0.7 and 0.6 ml min$^{-1}$. However, only a shift of the retention times was observed that shortened or extended the analysis time, respectively. Afterwards, the eluent concentrations

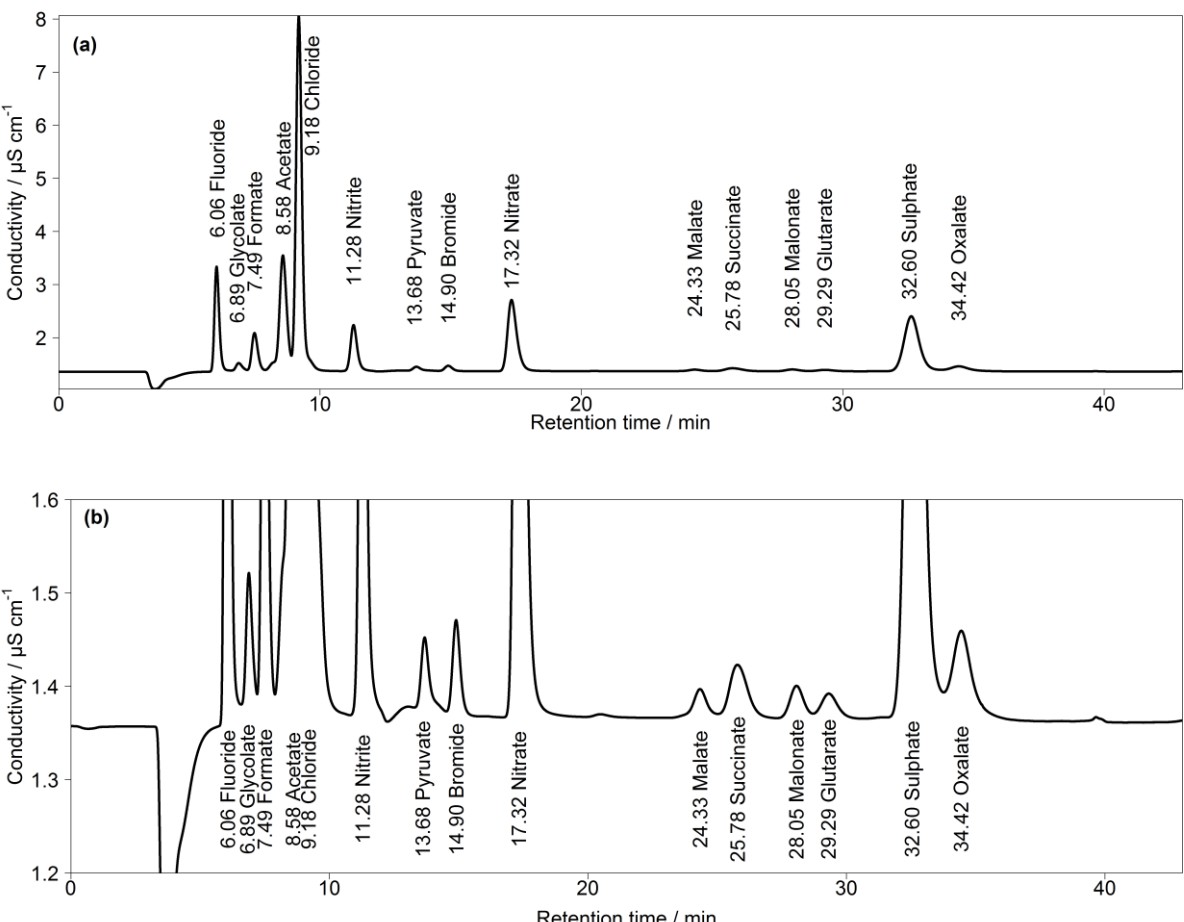

**Figure 2.** (a) First chromatogram of a standard solution with aqueous concentrations of 10 µg l$^{-1}$ for Cl$^-$, NO$_3^-$, SO$_4^{2-}$, 5 µg l$^{-1}$ for NO$_2^-$ and 1 µg l$^{-1}$ for F$^-$, Br$^-$ as well as all organic acids. Numbers in front of the ion names are the retention times. T = 65 °C and eluent flow of 0.8 ml min$^{-1}$. (b) Zoom in of (a).

were varied. An impact of NaOH in the eluent was not observed. The retention time shift was negligible and the separation was not affected. Detectable improvements were found for lower Na$_2$CO$_3$ concentrations between F$^-$ and Cl$^-$. The best separation for the MCAs was found for 6 mM Na$_2$CO$_3$. Here, glycolate and formate were baseline separated and the separation of acetate and Cl$^-$ was improved. However, the small peaks of the DCA were broadened, which could have a negative influence on the peak detection, especially for oxalate that is near the tail of SO$_4^{2-}$. An eluent composition of 8 mM Na$_2$CO$_3$ led to sharper DCA peaks, but worsened the MCA separation.

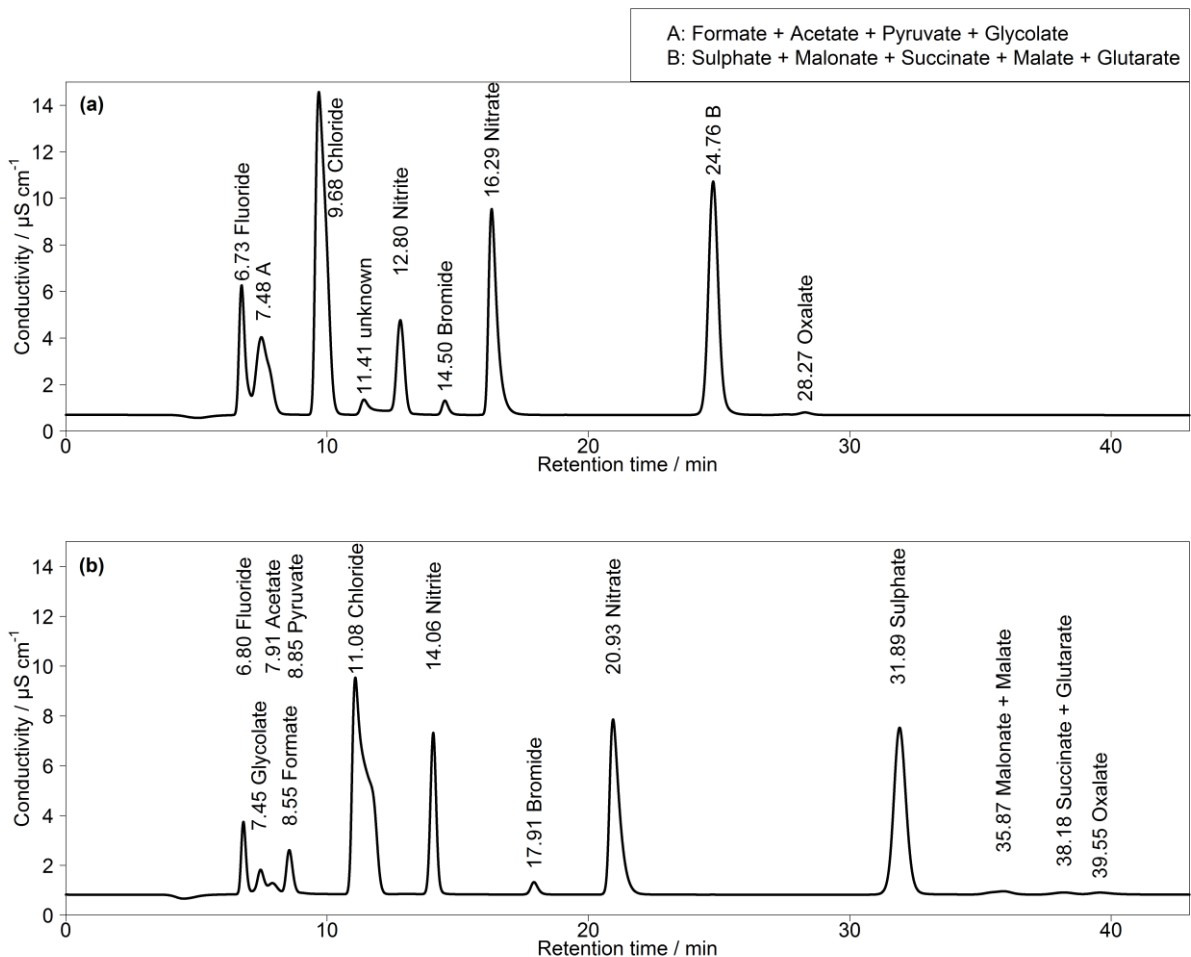

**Figure 3.** (a) Isocratic chromatogram for Shodex IC SI-50 4E with a standard solution of 50 µg l$^{-1}$ for Cl$^-$, NO$_3^-$, SO$_4^{2-}$, 25 µg l$^{-1}$ for NO$_2^-$ and 5 µg l$^{-1}$ for F$^-$, Br$^-$ as well as 2 µg l$^{-1}$ for all organic acids (T = 30 °C, eluent flow = 0.7 ml min$^{-1}$). (b) Isocratic chromatogram for Metrosep A Supp 7 with a standard solution of 50 µg l$^{-1}$ for Cl$^-$, NO$_3^-$, SO$_4^{2-}$, 25 µg l$^{-1}$ for NO$_2^-$ and 5 µg l$^{-1}$ for F$^-$, Br$^-$ as well as 3 µg l$^{-1}$ for all organic acids (T = 45 °C, eluent flow = 0.8 ml min$^{-1}$). Numbers in front of the ion names indicate the retention times in min.

To combine the advantages of the different eluent compositions, a gradient system was applied. Two different concentrated eluents were prepared. Within the Compact-IC, a highly concentrated eluent B (20 mM Na$_2$CO$_3$ and 0.75 mM NaOH) was mixed with a lower concentration eluent A (0.5 mM Na$_2$CO$_3$ and 0.75 mM NaOH). An example of the resulting chromatogram with the respective time program is shown in Fig. 4. In this example, the fraction of eluent B was increased to 50% shortly before the beginning of the analysis to shorten the analysis time before the F$^-$ peak eluted. At retention time t = 5 min eluent B was set to 0 %, which enabled a baseline-separation of the MCAs. At t = 15 min, eluent B was rapidly increased to 50 % to accelerate the analysis and, additionally, to obtain sharper peaks of SO$_4^{2-}$ and the DCAs. For subsequent analyses, it was

important to decrease eluent B to 0 % at t = 38 min. Otherwise, a shift of the retention times in the next analysis occurred because of a carryover of eluent B. Thus, the column was flushed with 100 % eluent A between analyses. The overall eluent profile is shown in Fig. 4a.

5  **Table 2.** Overview of the varied flows and eluent compositions in the isocratic system using the Metrosep A Supp 16 250 mm column with their effects on separation and reference to the corresponding figures in the supplement.

| | Flow | Eluent composition | Effect on separation | supplement |
|---|---|---|---|---|
| 1 | 0.8 ml min$^{-1}$ | 7 mM Na$_2$CO$_3$ / 0.75 mM NaOH | reference | |
| 2 | 0.6 ml min$^{-1}$ | 7 mM Na$_2$CO$_3$ / 0.75 mM NaOH | longer analysis time, shift of retentions times | S3 |
| 3 | 0.7 ml min$^{-1}$ | 7 mM Na$_2$CO$_3$ / 0.75 mM NaOH | longer analysis time, shift of retentions times | S4 |
| 4 | 0.9 ml min$^{-1}$ | 7 mM Na$_2$CO$_3$ / 0.75 mM NaOH | shorter analysis time, shift of retentions times | S5 |
| 5 | 1.0 ml min$^{-1}$ | 7 mM Na$_2$CO$_3$ / 0.75 mM NaOH | shorter analysis time, shift of retentions times | S6 |
| 6 | 0.8 ml min$^{-1}$ | 6 mM Na$_2$CO$_3$ / 0.75 mM NaOH | improved baseline-separation for MCA, broad DCAs | S7 |
| 7 | 0.8 ml min$^{-1}$ | 6.5 mM Na$_2$CO$_3$ / 0.75 mM NaOH | improved baseline-separation for MCA, broad DCAs | S8 |
| 8 | 0.8 ml min$^{-1}$ | 7.5 mM Na$_2$CO$_3$ / 0.75 mM NaOH | sharper DCA peaks, weaker MCA separation | S9 |
| 9 | 0.8 ml min$^{-1}$ | 8 mM Na$_2$CO$_3$ / 0.75 mM NaOH | sharper DCA peaks, weaker MCA separation | S10 |
| 10 | 0.8 ml min$^{-1}$ | 7 mM Na$_2$CO$_3$ / 0.65 mM NaOH | no improvements | S11 |
| 11 | 0.8 ml min$^{-1}$ | 7 mM Na$_2$CO$_3$ / 0.7 mM NaOH | no improvements | S12 |
| 12 | 0.8 ml min$^{-1}$ | 7 mM Na$_2$CO$_3$ / 0.8 mM NaOH | no improvements | S13 |
| 13 | 0.8 ml min$^{-1}$ | 7 mM Na$_2$CO$_3$ / 0.85 mM NaOH | no improvements | S14 |

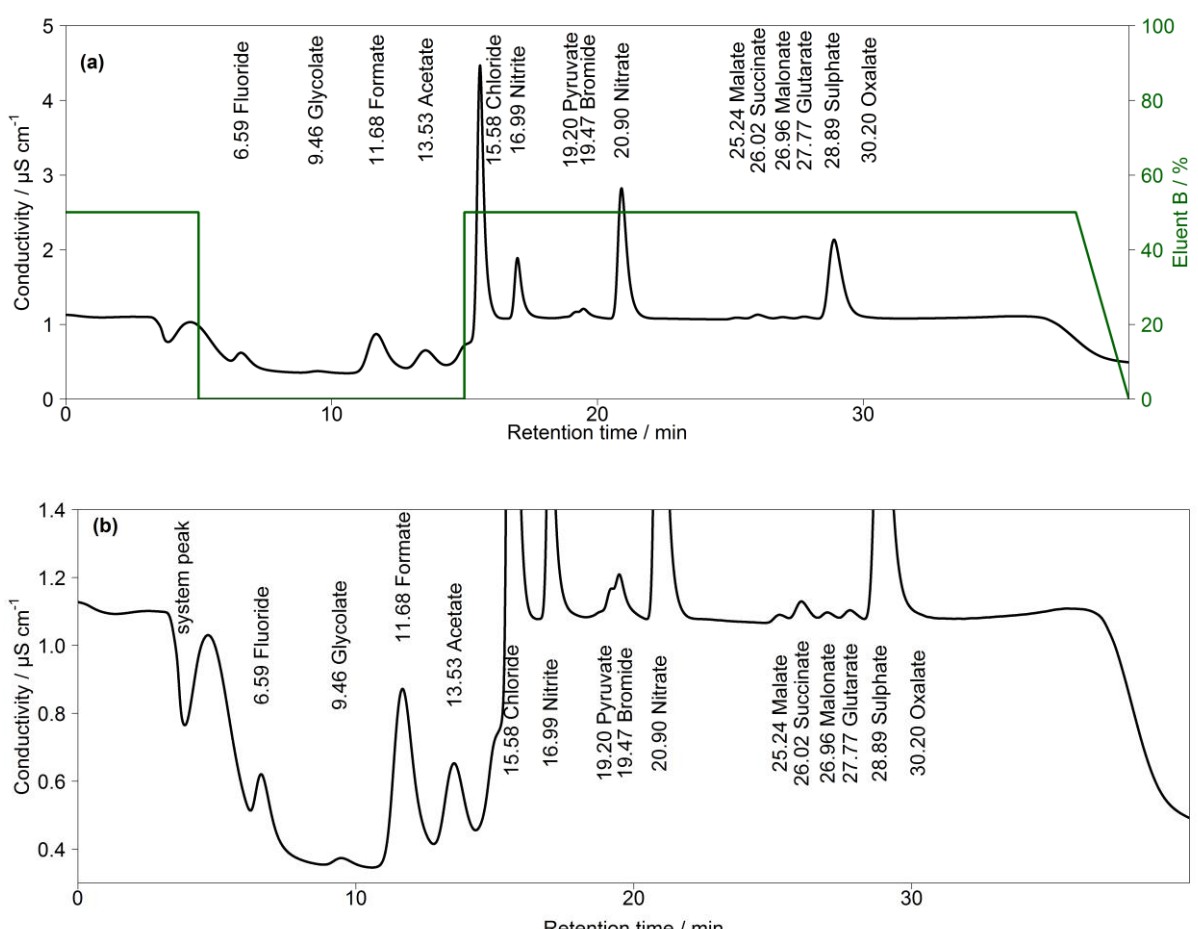

**Figure 4.** (a) Chromatogram of an aqueous standard solution with concentrations of 10 µg l$^{-1}$ for Cl$^-$, NO$_3^-$, SO$_4^{2-}$, 5 µg l$^{-1}$ for NO$_2^-$ and 1 µg l$^{-1}$ for F$^-$, Br$^-$ as well as all organic acids with the gradient system. Numbers in front of the ion names indicate the retention times in min. The fraction of eluent B within the eluent mixture over time is given as the green line. The eluent concentration of A is 0.5 mM Na$_2$CO$_3$ / 0.75 mM NaOH and of B is 20 mM Na$_2$CO$_3$ / 0.75 mM NaOH. T = 65 °C and eluent flow of 0.8 ml min$^{-1}$. (b) Zoom in of (a).

Although the MCA separation was improved by applying the described gradient, no baseline-separated pyruvate and Br$^-$ was observed. In addition, the DCA eluted very closely and oxalate co-eluted with the tail of the SO$_4^{2-}$ peak.

Changing the oven temperature was also considered. All measurements with the Metrosep A Supp 16 250 mm were executed with a temperature of 65 °C, which is the maximum temperature of the oven. Tests with 55 °C resulted in a shift of the retention times compared to an analysis with 65 °C, which is shown in Fig. S15. While the oxalate peak was sharper for the lower oven temperature, separation of the pyruvate and Br$^-$, as well as the other DCAs, was still not satisfactory.

To improve peak resolutions, the Metrosep A Supp 16 250 mm was extended with an additional Metrosep A Supp 16 150 mm (Metrohm, Switzerland) column. An even longer second column could not be chosen because of a system pressure limitation of 20 MPa that would otherwise be exceeded. Due to the increased back-pressure of the coupled columns, it was necessary to keep the oven temperature at 65 °C, because lower temperatures would increase the system pressure above its limit. The

chromatogram in Fig. 5 shows an improved separation with the extended column system.

The gradient profile was adjusted for this separation. First analyses were performed with the described profile of Fig. 4 but the retention times were not stable. The longer analysis time of 52.5 min and, thus, the shorter regeneration time between the analyses led to a carryover of eluent B. Therefore, other gradient profiles were tested and the best result was found for starting with 100 % of eluent A. Afterwards, eluent B was slowly increased to 40 % from t = 8 min until t = 25 min. This ensured a

proper separation of all MCAs. The concentration of eluent B was kept constant until t = 50 min, yielding an improved separation of DCAs as well as of $SO_4^{2-}$ and oxalate, even for higher $SO_4^{2-}$ concentrations. After t = 50 min, eluent B was decreased to 0 %. The overall eluent profile is shown in Fig. 5. The small peak behind acetate was identified as lactate. This ion was only detected in standard solutions and was not observed in ambient samples. Thus, a contamination from the chemicals or glassware used is likely.

The described method allowed for the proper separation of all organic target anions, which is why this system was selected and applied for real atmospheric analyses.

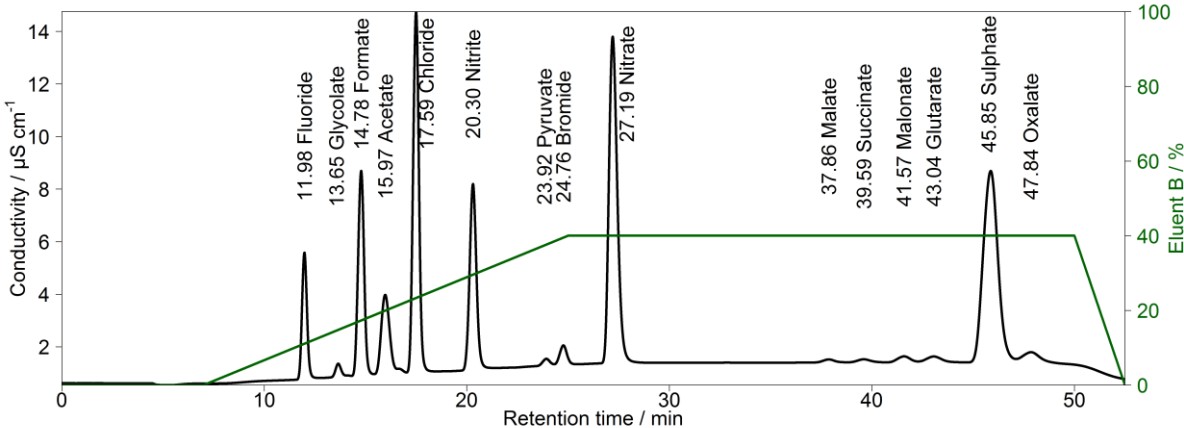

**Figure 5.** Chromatogram of combined Metrosep A Supp 16 250 mm and 150 mm column with a gradient eluent. The concentrations of the standard solution are 50 µg $l^{-1}$ for $Cl^-$, $NO_2^-$, Formate and Acetate; 100 µg $l^{-1}$ for $NO_3^-$, $SO_4^{2-}$; 10 µg $l^{-1}$

for and 10 µg $l^{-1}$ for $F^-$, $Br^-$ as well as all organic acids. Numbers in front of the ion names indicate the retention times in min. The fraction of eluent B within the eluent mixture over time is given as the green line. The eluent concentration of A is 0.5 mM $Na_2CO_3$ / 0.75 mM NaOH and of B is 20 mM $Na_2CO_3$ / 0.75 mM NaOH. T = 65 °C and eluent flow of 0.8 ml $min^{-1}$.

### 3.2 Limits of detection, linearity and precision

All values of the Limits of Detection (LOD), linearity and precision for each species are given in Table 3. The linearity of the calibration curve was determined after Funk et al. (2005). For the linear calibration function ($y = a + bx$), the slope $b$ and the intercept $a$ can be calculated as follows (DIN 32645, 2008):

$$Q_{xx} = \sum_{i=1}^{n} x_i^2 - \frac{\left(\sum_{i=1}^{n} x_i\right)^2}{n} \tag{1}$$

$$Q_{yy} = \sum_{i=1}^{n} y_i^2 - \frac{\left(\sum_{i=1}^{n} y_i\right)^2}{n} \tag{2}$$

$$Q_{xy} = \sum_{i=1}^{n} (x_i y_i) - \frac{\sum_{i=1}^{n} x_i \sum_{i=1}^{n} y_i}{n} \tag{3}$$

$$b = \frac{Q_{xy}}{Q_{xx}} \tag{4}$$

$$a = \bar{y} - b\bar{x} \tag{5}$$

where $Q_{xx}$, $Q_{yy}$ and $Q_{xy}$ are the square sums, $\bar{x}$ and $\bar{y}$ the means and $n$ the number of calibration points.

The calibration of a non-linear second-order function ($y = a + bx + cx^2$) was calculated considering DIN ISO 8466-2 (2004). Simultaneous to the Eq. (1) to (3), the following quadratic sums were added:

$$Q_{x^3} = \sum_{i=1}^{n} x_i^3 - \sum_{i=1}^{n} x_i \frac{\sum_{i=1}^{n} x_i^2}{n} \tag{6}$$

$$Q_{x^4} = \sum_{i=1}^{n} x_i^4 - \frac{\left(\sum_{i=1}^{n} x_i^2\right)^2}{n} \tag{7}$$

$$Q_{x^2 y} = \sum_{i=1}^{n} (x_i^2 y_i) - \sum_{i=1}^{n} y_i \frac{\sum_{i=1}^{n} x_i^2}{n} \tag{8}$$

The intercept $a$ and the coefficients $b$ and $c$ were calculated as follows:

$$c = \frac{Q_{xy}Q_{x^3} - Q_{x^2y}Q_{xx}}{Q_{x^3}^2 - Q_{xx}Q_{x^4}} \tag{9}$$

$$b = \frac{Q_{xy} - cQ_{x^3}}{Q_{xx}} \tag{10}$$

$$a = \bar{y} - b\bar{x} - c\frac{\sum_{i=1}^{n} x_i^2}{n} \tag{11}$$

The residual standard deviation for the linear $s_{y,l}$ and the non-linear case $s_{y,nl}$ are:

$$s_{y,l} = \sqrt{\frac{\sum_{i=1}^{n}[y_i - (bx_i + a)]^2}{n-2}} = \sqrt{\frac{Q_{yy} - \frac{Q_{xy}^2}{Q_{xx}}}{n-2}} \tag{12}$$

$$s_{y,nl} = \sqrt{\frac{\sum_{i=1}^{n}(y_i - \hat{y}_i)^2}{n-3}} \tag{13}$$

To test each ion's linearity, the difference of the variances $DS^2$ was calculated after Funk et al. (2005):

$$DS^2 = (n-2)s_{y,l}^2 - (n-3)s_{y,nl}^2 \tag{14}$$

with the degree of freedom of $f = 1$. For a F-Test, the test value $TV$ was determined:

$$TV = \frac{DS^2}{s_{y,nl}^2} \tag{15}$$

This test value was compared with a F-Test table ($f_1 = 1$, $f_2 = n - 3$, $P = 99$ %). If $TV \leq F$, the calibration function is linear. For the other cases, the calibration function is a non-linear second-order function. In the case of the present work $F = 11.26$. The resulting $TV$ values for each ion are summarized in Table S1. Depending on the result of the linearity test, linear or

20   quadratic calibration functions were fitted. As examples for a linear and a quadratic fit, the calibration functions of $NO_3^-$ and

pyruvate are respectively displayed in Fig. 6. Other calibration functions are given in the supplement (Fig. S16). The linearity test was performed through a double injection of 11 standards with evenly distributed concentrations over one order of magnitude, where the maximum concentration corresponded to standard 3 in Table 1.

The Limits of Detection (LOD) for the Compact-IC were estimated from mean blank values plus three times the standard deviation ($3\sigma$). For species that were not found in the blank measurements, the LOD represents the smallest observable peak. The LODs as atmospheric concentrations varied between 0.5 ng m$^{-3}$ for malonate and 17.4 ng m$^{-3}$ for glutarate and were considered sufficiently low for field application of the system. The precision of the method was calculated as the relative standard deviation (RSD) of the peak area of 11 injections of standard 3 (Table 1) over one month. For all ions, the precision is below 3 %, indicating a good repeatability.

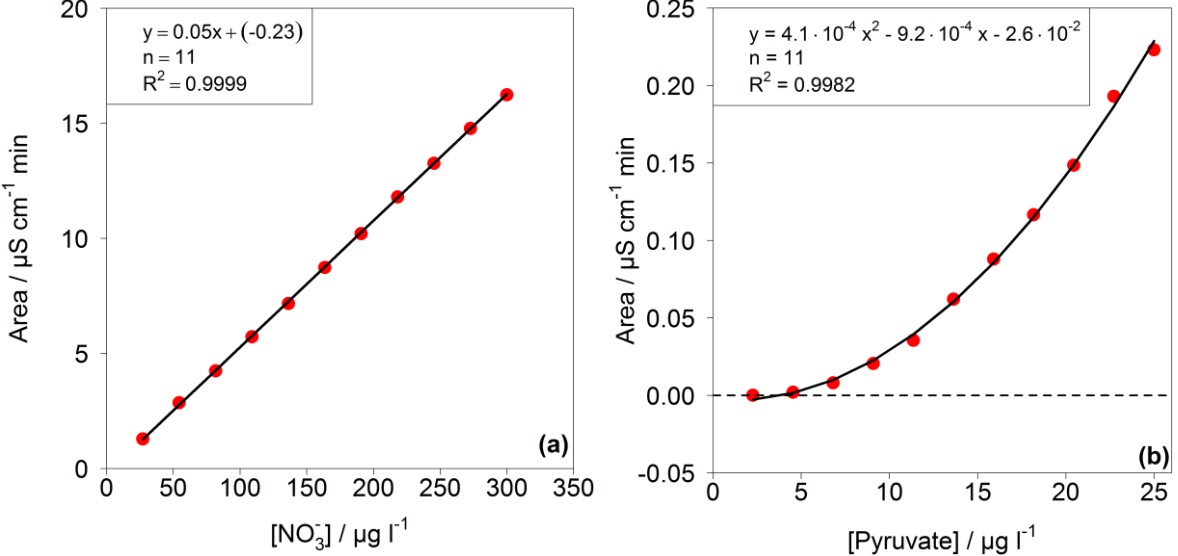

**Figure 6.** (a) Linear (NO$_3$$^-$) and (b) quadratic (pyruvate) calibration function.

**Table 3.** Type of calibration curve, LODs and the method precision for each ion.

| Ion | Cal. curve | LOD / ng m$^{-3}$ | precision / % |
|---|---|---|---|
| F$^-$ | quadratic | 4.4 | 0.3 |
| Cl$^-$ | linear | 16.2 | 1.9 |
| NO$_2^-$ | linear | 2.3 | 0.4 |
| Br$^-$ | quadratic | 17.0 | 0.2 |
| NO$_3^-$ | linear | 5.4 | 0.7 |
| SO$_4^{2-}$ | quadratic | 5.5 | 0.4 |
| Methanesulfonate | quadratic | 1.3 | 0.5 |
| Formate | linear | 6.2 | 0.5 |
| Acetate | quadratic | 3.9 | 1.0 |
| Glycolate | quadratic | 3.8 | 2.9 |
| Propionate | linear | 12.5 | 0.7 |
| Butyrate | linear | 16.0 | 0.5 |
| Pyruvate | quadratic | 13.4 | 0.1 |
| Oxalate | linear | 1.4 | 0.1 |
| Malonate | linear | 0.5 | 0.1 |
| Malate | quadratic | 3.6 | 0.2 |
| Succinate | quadratic | 11.7 | 1.0 |
| Glutarate | quadratic | 17.4 | 0.1 |

## 3.3 Sample handling

For the combination of the MARGA and the Compact-IC, the liquid flows in the system had to be adjusted to achieve a high time resolution and to analyze the solutions as fast as possible after the sampling. As an overview, a schematic setup and a

5   time diagram in Fig. 7 displays the important steps for the CA analysis of the WRD and SJAC samples. Therein, the sampled airflow is described with green arrows. Syringe pumps within the MARGA collected the dissolved ions within the WRD (blue arrows) and SJAC (red arrows) solutions. This sampling required one hour and yields 25 ml of sample solution in each of the two syringes.

In the second hour, the MARGA syringe pumps transported the solutions to the IC system within the MARGA to analyze the

10  inorganic compounds in the gas and particle-phase, as well as to the autosampler of the Compact-IC. Thereby, the WRD solution was injected with a flow of 0.417 ml min$^{-1}$ into the MARGA-IC to rinse the sampling lines and to fill the injection loop for the first 13 minutes. Afterwards the analysis of this sample followed for 17 minutes. In the second 30-minutes interval, the SJAC sample was injected and analyzed. Only during the injections into the MARGA-IC of both the WRD and SJAC

samples, no solutions were transported via an external 6-way-valve (Fig. 1 (g)) either to the autosampler or to the waste. As the vials in the autosampler had a volume of 12.5 ml, the 6-way-valve transferred the samples from the WRD and SJAC to the autosampler only for the first 45 minutes and the rest of the solutions were directed to waste. In the third hour, the WRD sample was pre-concentrated and was analyzed by the Compact IC. Afterwards, the SJAC sample was pre-concentrated and was analyzed in the fourth hour.

To achieve a pre-concentration and analysis of one sample in one hour, the transfer of analytes from the autosampler to the Compact-IC and the pre-concentration of the sample had to be performed within the remaining 7.5 minutes, as the final Compact-IC analysis described previously needed 52.5 minutes. Therefore, the sample flows were increased to 4 ml min$^{-1}$, which is the maximum for what is allowed for the pre-concentration column. For the quantification of the organic acids with the Compact-IC, the hourly integrated MARGA samples were collected every two hours.

In the following, the MARGA and the Compact-IC analysis will be distinguished. Analyzed ions by the MARGA were measured by the original MARGA system, while ions from the Compact-IC were measured by the added setup.

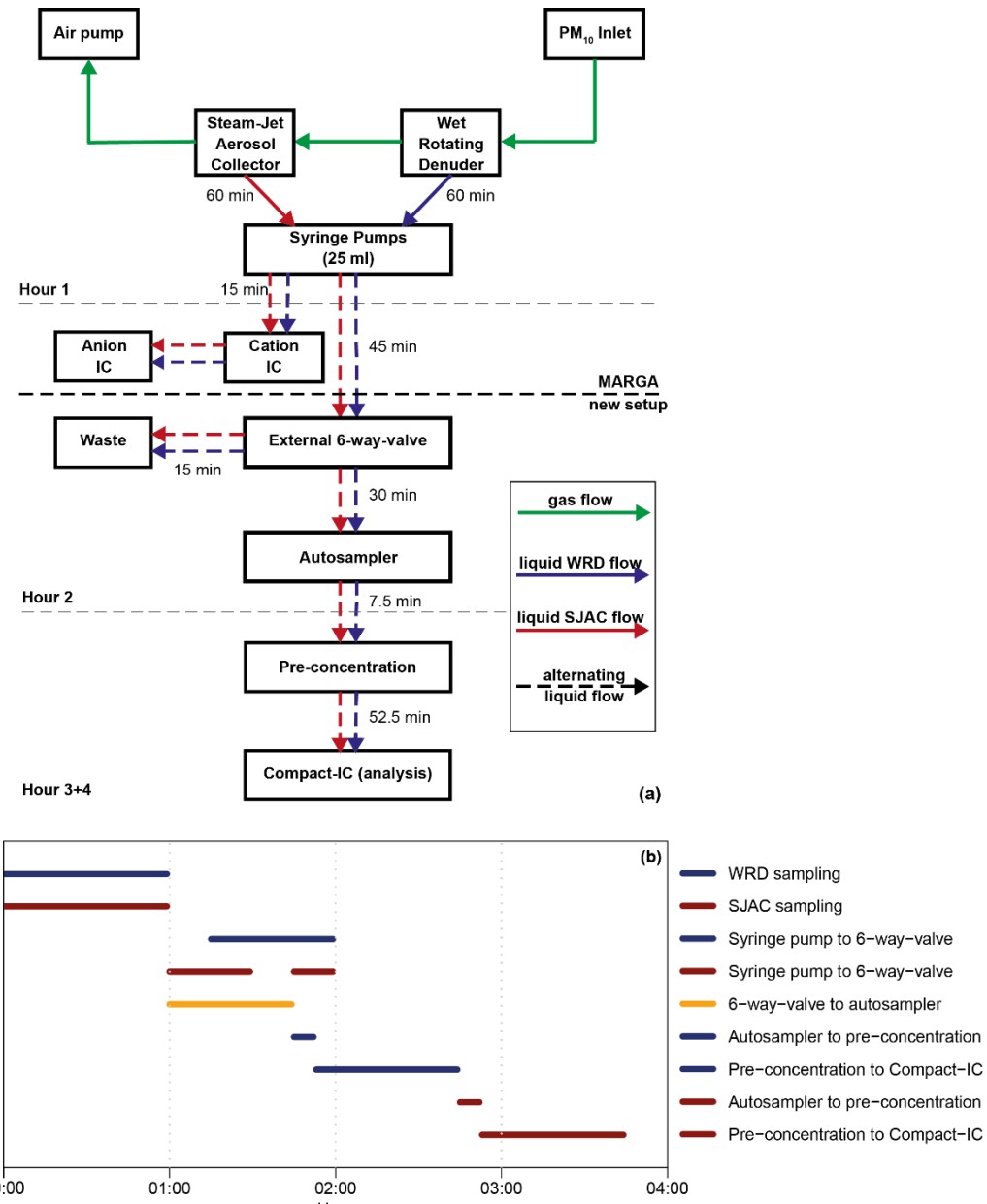

**Figure 7.** (a) Schematic overview of the sample handling for the complete setup (MARGA, Autosampler and Compact-IC). Green arrows illustrate gas flow, while blue and red are the aqueous WRD and SJAC samples, respectively. The time on the arrows represents the flow duration. Dashed arrows stand for alternating flows. Grey dashed lines illustrate the hourly time steps, while the black bold dashed line is the border between MARGA and the Compact-IC. (b) Time diagram of the liquid handling (blue: WRD, red: SJAC). Black is the open mode of the 6-way-valve for liquid transport from the syringe pumps to the autosampler.

### 3.4 MARGA absorption solution

The original MARGA absorption solution in the denuder and SJAC contains 10 mg l$^{-1}$ H$_2$O$_2$ to avoid biological contamination of the system and to oxidize absorbed SO$_2$ into SO$_4^{2-}$. However, H$_2$O$_2$ can affect the concentration of dissolved CAs through oxidation. As an example, Schöne and Herrmann (2014) described a fast degradation of pyruvate in an aqueous H$_2$O$_2$ solution and a simultaneous increase of acetate as a product. This would result in incorrect concentrations within the aqueous gas and particle-phase samples. Therefore, H$_2$O$_2$ was not added during the CA measurements and ultrapure water was used as the absorption solution. With the missing oxidant, however, an underestimation of measured MARGA SO$_2$ occurred that required further adjustments. A MARGA software update by Metrohm-Applikon (the Netherlands) allowed for integration of the sulphite peak and calculating the overall SO$_2$ concentration as a sum of sulphite and SO$_4^{2-}$. For the Compact-IC analysis, the sulphite peak was located between glutarate and SO$_4^{2-}$ and did not interfere with the quantification of the organic acids. Because of the missing biocide H$_2$O$_2$, the MARGA was cleaned more frequently, at least every two weeks to avoid bacterial contamination. The WRD and SJAC were rinsed with ethanol as well as ultrapure water and afterwards a system cleaning procedure was applied. Therein, the absorption solution was replaced by a 1 % H$_2$O$_2$ solution with maximal syringe pump speed for at least three hours. Before the next MARGA analysis, the complete MARGA system was rinsed for two hours with ultrapure water to remove the H$_2$O$_2$.

### 3.5 WRD efficiency and WRD particle collection

When measuring the gas and particle-phase with a combination of WRD and SJAC, the collection efficiency of gases and the particle penetration within the denuder should be investigated. In the literature, experimentally derived collection efficiencies are available for annular denuders that correspond with the WRD within the MARGA. Wyers et al. (1993) published an NH$_3$ efficiency of 98% with an air flow of 30 l min$^{-1}$. Khlystov et al. (2009) investigated the HNO$_3$ breakthrough in the presence of particles. They found under urban background conditions a breakthrough of 0.6%. The MARGA manufacturer Metrohm-Applikon already tested the denuder efficiency of the WRD for SO$_2$ and found a recovery of 99.7% (personal communication). In the present study, the collection efficiencies of the annular WRD were theoretically calculated for the different inorganic and organic acids following different approaches suggested in the literature (Possanzini et al., 1983;Winiwarter, 1989;De Santis, 1994;Berg et al., 2010). For each approach, all equations for the denuder efficiency calculation are given in the supplement. Calculated denuder efficiencies are summarized in Table 4. The calculated efficiencies according to Possanzini et al. (1983), De Santis (1994) and Berg et al. (2010) are higher than 99% indicating a nearly complete absorption of the investigated gases within the WRD. Denuder efficiencies derived from the approach of Winiwarter (1989) ranges between 95% for glutaric acid and nearly 100% for the inorganic gases as well as formic acid. However, compared to the other three studies, Winiwarter (1989) did not consider the geometry of an annular denuder in his approach and is therefore considered to be less accurate. Regarding the efficiencies calculated after Possanzini et al. (1983) and the highest formic acid gas-phase

concentrations of 7.58 µg m$^{-3}$ measured by the Compact-IC, the potential particulate artifact concentration would result in 2.3 ng m$^{-3}$, which is below the LOD of the method. The calculated WRD efficiencies of Possanzini et al. (1983) were used to calculate the potential particulate artifact concentrations from the gaseous concentrations for the complete measurement campaign. These concentrations were compared with real particulate concentrations. It was found that the artifact concentration

is far below the real measurements. Only an average of 0.15% of the real particulate formate concentrations could be explained by possible formic acid breakthrough. For acetate, propionate, butyrate, pyruvate and glycolate, percental values of 3.7%, 0.6%, 0.3%, 0.4% and 0.2% were calculated, respectively, and were similar low. Thus, interferences of gaseous compounds in the SJAC are negligible.

Another method to evaluate the WRD collection efficiency of gases is the comparison of measured compounds that are

predominantly found in the gas phase. The inorganic nitrite ($NO_2^-$) of HONO is quantified by the Compact-IC. The $NO_2^-$ concentration within the particle-phase is near zero. For the three highest HONO concentrations during the measurement campaign (on average 5.1 µg m$^{-3}$) only 66.2 ng m$^{-3}$ particle-phase $NO_2^-$ were observed resulting in a maximum HONO breakthrough of 1.3%. The same calculation were performed for formic and acetic acid that are most abundant in the gas-phase (Nah et al., 2018a), resulting in a potential maximal breakthrough of 0.7% and 0.1%, respectively. Thus, the calculated denuder

efficiencies are in same range with the experimental derived ones reported in the literature.

For the WRD particle collection, Wyers et al. (1993) investigated possible particulate ammonium collection within the denuder. A sampling of ammonium sulphate particles of 0.1 and 1 µm median volume diameter resulted in a particle collection of 0.6% within the denuder for both sizes. In the same range are the experimentally derived particle collections of Possanzini et al. (1983) with 0.2% for 0.3-0.5 µm particles and 1.4% for particles larger than 3 µm.

The three highest concentrations of the DCAs were compared with the gas-phase concentrations measured in the field. The oxalate concentrations in the particle-phase ranged between 327 ng m$^{-3}$ and 543 ng m$^{-3}$. At the same time, no gas-phase concentrations were detected. For the other particulate dicarboxylic acids and methanesulfonate, detectable gas-phase concentrations were not observed during periods with high particulate concentrations indicating a negligible effect of particle collection within the WRD.

In conclusion, the calculated denuder efficiencies that are in agreement with the literature as well as low rates of denuder breakthrough and low particulate losses within the WRD approve the use of a coupled WRD/SJAC system as valid method to separate the gas and particle-phases for the sampling of the low-molecular weight organic acids.

**Table 4.** Diffusion coefficients (D) calculated according to Fuller et al. (1966) and calculated annular denuder efficiencies (E) according to the equations of Winiwarter (1989), Possanzini et al. (1983), De Santis (1994) and Berg et al. (2010) for gases.

| Gas | $D$ / cm$^2$ s$^{-1}$ | $E_{Winiwarter}$ / % | $E_{Possanzini}$ / % | $E_{De\ Santis}$ / % | $E_{Berg}$ / % |
|---|---|---|---|---|---|
| HCl | 0.163 | 99.75 | 99.99 | >99.99 | >99.99 |
| HONO | 0.171 | 99.81 | 99.99 | >99.99 | >99.99 |
| HNO$_3$ | 0.157 | 99.69 | 99.99 | >99.99 | >99.99 |
| SO$_2$ | 0.149 | 99.58 | 99.98 | >99.99 | >99.99 |
| NH$_3$ | 0.199 | 99.93 | >99.99 | >99.99 | >99.99 |
| Formic acid | 0.143 | 99.48 | 99.97 | >99.99 | >99.99 |
| Acetic acid | 0.119 | 98.76 | 99.89 | 99.99 | 99.98 |
| Propionic acid | 0.104 | 97.87 | 99.74 | 99.96 | 99.95 |
| Butyric acid | 0.094 | 96.95 | 99.55 | 99.91 | 99.90 |
| Pyruvic acid | 0.103 | 97.80 | 99.72 | 99.96 | 99.94 |
| Glycolic acid | 0.114 | 98.52 | 99.85 | 99.98 | 99.97 |
| Oxalic acid | 0.113 | 98.46 | 99.84 | 99.98 | 99.97 |
| Malonic acid | 0.100 | 97.54 | 99.68 | 99.94 | 99.93 |
| Succinic acid | 0.0912 | 96.62 | 99.47 | 99.89 | 99.87 |
| Malic acid | 0.0892 | 96.37 | 99.41 | 99.88 | 99.85 |
| Glutaric acid | 0.0827 | 95.41 | 99.15 | 99.80 | 99.77 |
| Methanesulfonic acid | 0.114 | 98.52 | 99.85 | 99.98 | 99.97 |

## 3.6 Intercomparison of inorganic ions

5   Both the MARGA and Compact-IC determined the inorganic compounds in the gas and particle-phase from the same aqueous solution but used different IC methods, including a different calibration, different sample enrichments, different separation columns and different eluent compositions and profiles. For quality assurance, the inorganic ions were compared in the gas and particle-phase for the complete one year field application of the extended MARGA system and the results are summarized numerically in Table 5 as well as graphically in Fig. S17 and S18.

10   The Cl$^-$, NO$_3^-$ and SO$_4^{2-}$ concentrations measured by the MARGA and the Compact-IC were in good agreement with $R^2 = 0.95$, $R^2 = 0.97$ and $R^2 = 0.99$, respectively, and slopes close to unity. It should be noted that the MARGA measured the overall SO$_2$ concentration because of the quantification of sulphite and SO$_4^{2-}$, while the Compact-IC quantified only SO$_4^{2-}$ and not the sulphite peak. Thus, higher MARGA SO$_2$ concentrations were expected. An underestimation of the SO$_2$ concentration by the

Compact-IC was indeed obvious, especially for lower concentrations. However, the overall correlation for $SO_2$ was found to be good with a slope of 0.98 and a $R^2$ of 0.97. It is likely that the WRD absorbed gaseous atmospheric oxidants that oxidized sulphite into $SO_4^{2-}$ in the aqueous WRD solution, even without the addition of $H_2O_2$ to the absorption solution, as described above. With slopes around 1.5, the regression parameters of HCl and $HNO_3$ are comparable with each other. The slope for the

HCl comparison is a result of three MARGA outliers with concentrations higher than 4 µg m$^{-3}$. Without these outliers the slope decreased to 1.16 and the coefficient of determination to $R^2 = 0.79$. A decrease from $R^2 = 0.77$ to $R^2 = 0.57$ was observed when $HNO_3$ concentrations above 3 µg m$^{-3}$ were removed. The elimination of the outliers did not result in an improvement of the slope. In the case of $HNO_3$ and partly of HCl, the MARGA quantified higher concentrations for the same aqueous solution. Rumsey and Walker (2016) found a quadratic response for low $HNO_3$ concentrations and hypothesized an overestimation of

MARGA concentrations. The same was possible for HCl. In the present study, the particulate concentrations were higher and the quadratic influence is of minor importance, leading to slopes near unity.

**Table 5.** Orthogonal regression parameters of the comparison of inorganic compounds measured by the MARGA and Compact-IC in Melpitz for one year. Scatter plots are given in Fig. S17 and S18.

| Phase | Ion | Slope | Intercept | $R^2$ | n |
|---|---|---|---|---|---|
| gas | HCl | 1.50 | -0.08 | 0.92 | 1358 |
| | HONO | 0.80 | 0.20 | 0.59 | 2713 |
| | $SO_2$ | 0.98 | 0.12 | 0.97 | 2558 |
| | $HNO_3$ | 1.51 | 0.00 | 0.76 | 2570 |
| particle | $Cl^-$ | 1.11 | -0.02 | 0.95 | 1768 |
| | $NO_3^-$ | 1.02 | 0.20 | 0.99 | 2707 |
| | $SO_4^{2-}$ | 0.88 | 0.39 | 0.97 | 2705 |

The HONO comparison revealed an obvious scattering ($R^2 = 0.59$). Possible reactions between sampling and analysis altered the HONO concentrations. Spindler et al. (2003) observed and quantified the artifact sulphate and HONO formation by reactions of dissolved $NO_2$ and $SO_2$ within the aqueous solution of the WRD. Between the MARGA and the Compact-IC analysis for the WRD samples one hour passes where such artifact formation could occur. However, the intercomparison of

the more stable inorganic ions demonstrated a good comparability between the MARGA and Compact-IC data.

**3.7 Example application in the field**

To prove the suitability of the complete setup, two weeks of the one year measurement campaign are presented. Figure 8 displays the measured organic acids in the gas and particle-phase from 3$^{rd}$ May 2017 until 14$^{th}$ May 2017. Included grey shaded periods display downtimes of both the MARGA and the Compact-IC because of MARGA cleaning procedure (12$^{th}$ May),

blank measurements of the complete new MARGA setup ($12^{th}$ May) or measurements of calibration standards ($5^{th}$, $9^{th}$, $12^{th}$ May). Table 6 gives the percental data coverage, i.e. concentrations above LOD, for each organic acid in the gas and particle-phase during the uptime periods.

Very good data coverages were found for formate and acetate in both phases as well for glycolate and methanesulfonate in the particle-phase with percental values over 90%. Table 6 indicates the dominance of non-glycolate MCAs in the gas phase while DCAs were predominantly detected in the particle-phase. This finding is in agreement with the higher vapour pressures of MCAs (Howard and Meylan, 1997).

For the calculations of mean concentrations, all values below LOD were included and not detected data were set to zero. Mean (maximum) concentrations of 306 ng m$^{-3}$ (2207 ng m$^{-3}$) were observed for gaseous acetic acid followed by formic 199 ng m$^{-3}$ (919 ng m$^{-3}$), propionic 83 ng m$^{-3}$ (524 ng m$^{-3}$), pyruvic 76 ng m$^{-3}$ (253 ng m$^{-3}$), butyric 34 ng m$^{-3}$ (343 ng m$^{-3}$) and glycolic acid 32 ng m$^{-3}$ (259 ng m$^{-3}$). This is in agreement with other studies. Fisseha et al. (2006) monitored in the city of Zurich, Switzerland, mean concentrations of acetic acid between 1.09 µg m$^{-3}$ in September and 1.97 µg m$^{-3}$ in March. Formic and propionic acid ranged between 0.24 µg m$^{-3}$ and 1.07 µg m$^{-3}$ as well as between 0.16 µg m$^{-3}$ and 0.03 µg m$^{-3}$, respectively. Another urban site is described by Lee et al. (2009). In Seoul, they reported formic, acetic and propionic acid concentrations of around 3.83 µg m$^{-3}$, 4.99 µg m$^{-3}$ and 1.54 µg m$^{-3}$, respectively. Higher concentrations of formic and acetic acid at urban sites could be favored due to anthropogenic emissions. Khare et al. (1999) and references therein summarized formic and acetic acid concentrations and reported concentrations of 1.72 µg m$^{-3}$ and 2.25 µg m$^{-3}$ at a semiurban site in Central Germany, respectively. The amount of formic and acetic acid measured at the semiurban site Bondville, United States, are around 0.6 µg m$^{-3}$ and 1 µg m$^{-3}$, respectively (Ullah et al., 2006). Nah et al. (2018a) detected at the rural Yorkville, Georgia, site averaged concentrations of formic and acetic acid of 2.2 µg m$^{-3}$ and 1.9 µg m$^{-3}$, respectively

DCAs and methanesulfonate were rarely detected in the gas-phase due to the low vapour pressures of these compounds. Thus, an existence of these species is more likely in the particle phase. However, malonate, succinate, malate and glutarate were rarely or not at all detected in the particulate phase (Table 6). Oxalate is the predominant DCA in the particle phase with a percental data coverage of 77.3%. Interestingly, also formic and acetic acid were detected in the particulate phase. Mean (maximum) concentrations of 31 ng m$^{-3}$ (209 ng m$^{-3}$), 30 ng m$^{-3}$ (465 ng m$^{-3}$), 34 ng m$^{-3}$ (282 ng m$^{-3}$), 26 ng m$^{-3}$ (162 ng m$^{-3}$) and 18 ng m$^{-3}$ (54 ng m$^{-3}$) were monitored for oxalate, methanesulfonate, formate, glycolate and acetate, respectively.

As comparison, Parworth et al. (2017) detected average glycolate concentrations of 26.7 ng m$^{-3}$ with a nighttime maximum of around 60 ng m$^{-3}$ in Fresno during winter. Mean concentrations of particulate oxalate, acetate and formate of 0.07 µg m$^{-3}$, 0.06 µg m$^{-3}$ and 0.05 µg m$^{-3}$ were measured by Nah et al. (2018a), respectively. van Pinxteren et al. (2014) presented oxalate concentrations measured by impactors in Melpitz during autumn. A mean concentration of 52 ng m$^{-3}$ was published. Additionally, mean values measured in Falkenberg (approximately 25 km northeast of Melpitz) during summer peaks in 80 ng m$^{-3}$. A predominant peak of methanesulfonate was observed on $8^{th}$ May 2017. This happened simultaneously with a sudden increase of wind speed (Figure S19). Northwesterly winds transporting marine air masses to Melpitz appear to be the

most likely explanation as marine DMS oxidizes within the aqueous phase to methanesulfonate (Hoffmann et al., 2016;Lana et al., 2011).

The comparison with the literature shows rather low concentrations of the organic acids in the particle-phase during the example application in the field. A possible reason are the changeable weather conditions. The temperature varied during the first seven days between 0°C and 15°C. Afterwards, sunnier and warmer conditions were present resulting in an increase of formic, acetic and propionic acid concentrations probably because of a relationship between temperature, global radiation and higher organic acid concentration.

Highest values for temperature and global radiation were observed during daytime (Figure S19) why it is expected to measure elevated concentrations in this time. Diurnal cycles of formic acid and particulate oxalate are illustrated in Fig. 9a,b, respectively. Both compounds had the lowest concentrations in the early morning and increased in the afternoon until the maxima were reached in the evening following the observed average temperature. This trend is in agreement with previous studies (Lee et al., 2009;Millet et al., 2015;Khare et al., 1997;Nah et al., 2018b;Martin et al., 1991). During night, concentrations decreases due to deposition processes. Simultaneously, a decreasing surface temperature cools down the lower air layers leading to an inversion layer that suppresses the vertical mixing. The increasing concentrations after sunrise are likely a result of downward mixing of enriched layers above the boundary layer (Khare et al., 1999). Biogenic emissions and photochemical processes lead to increasing concentrations during daytime (Khare et al., 1999;Liu et al., 2012b).

Nah et al. (2018a) reached the same conclusion from their study. They found higher concentrations during warm and sunny days caused by elevated availability of biogenic precursors. During their two month measurement campaign in late summer and autumn, the maximum temperature ranged in average between 25°C and 30°C with high solar irradiances. Interestingly, for several days the temperature decreased below 20°C with a simultaneous decrease of several DCA concentrations. Thus, the low concentration found during the example application are probably a result of lower temperatures, low global radiation and, probably, lower amounts of biogenic precursors in spring.

The application in the field demonstrate the suitability of the developed setup. The measured concentrations of low-molecular weight organic acids in Melpitz are partly lower than concentrations, which can be found in the literature. The increase of the concentrations after 11th May 2017 indicate an influence of the increasing temperature and the available sunlight, which is needed for biogenic emissions or photochemical reactions of atmospheric precursors. Further in-depth analyses and detailed results of the one year measurements with the extended MARGA system will be presented elsewhere (Stieger et al., manuscript in preparation).

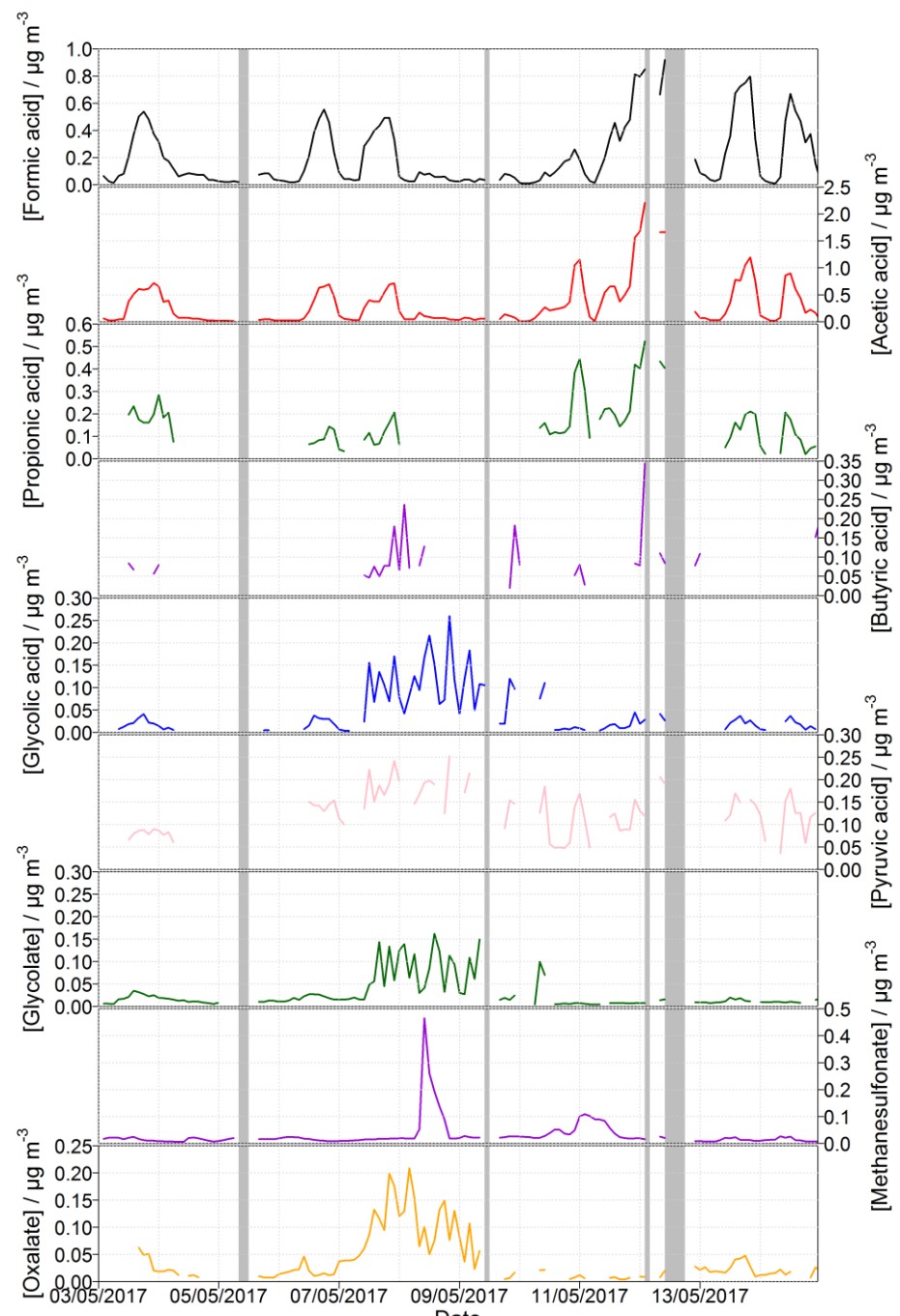

**Figure 8.** Measured concentrations for gaseous formic, acetic, propionic, butyric, glycolic and pyruvic acid as well as for particulate glycolate, methanesulfonate and oxalate from 3<sup>rd</sup> May until 14<sup>th</sup> May 2017 for an example application. The grey shaded areas represent periods without data because of instrumental issues.

**Table 6.** Data coverage for the organic species measured in the gas and particle phase from 3rd May 2017 until 14th May 2017 during instrument uptime periods shown for data above the LOD.

| Compound | Gas-phase / % | Particle-phase / % |
|---|---|---|
| Formate | 100 | 99.2 |
| Acetate | 99.2 | 96.2 |
| Propionate | 51.9 | 0.8 |
| Butyrate | 30.8 | 6.1 |
| Pyruvate | 58.6 | 5.3 |
| Glycolate | 70.7 | 90.9 |
| Oxalate | 25.6 | 77.3 |
| Malonate | 2.3 | 23.5 |
| Succinate | 3.8 | 9.1 |
| Malate | 1.5 | 27.3 |
| Glutarate | 0 | 0 |
| Methanesulfonate | 24.1 | 99.2 |

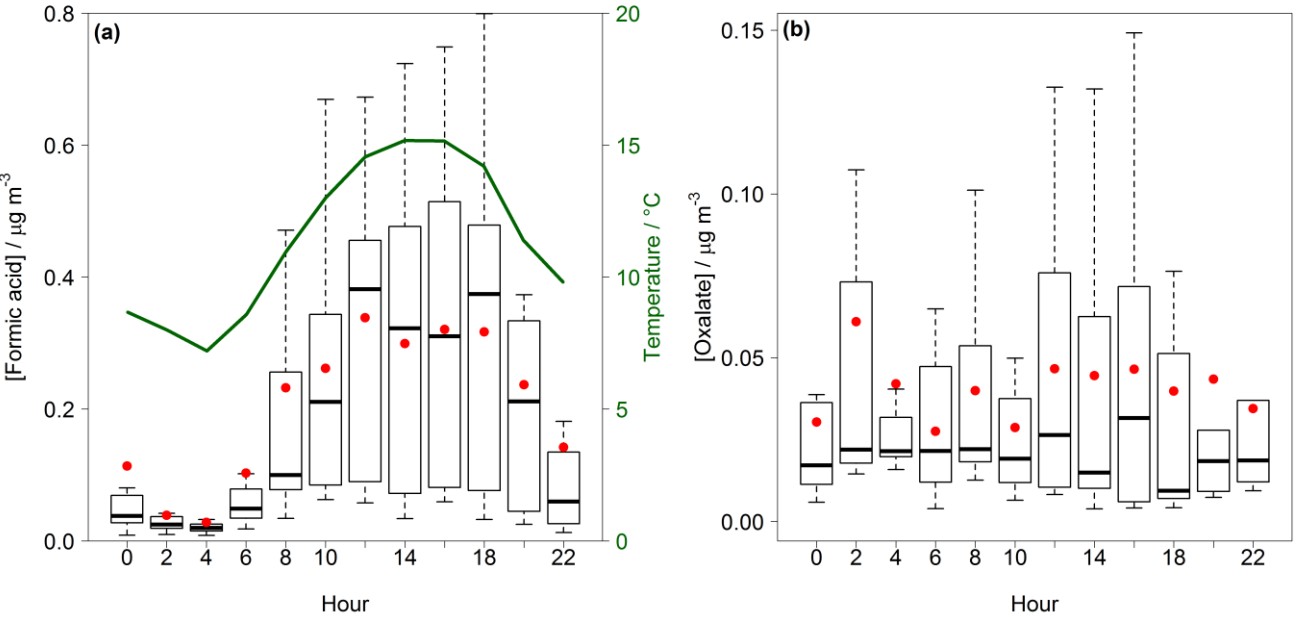

**Figure 9.** Box-Whisker-Plot for the diurnal variation of gaseous formic acid (a) and particulate oxalate (b) in Melpitz for the example application in the field. The red dots represent the mean, the box the 25th and 75th percentile and the upper whisker the 75th percentile plus the 1.5-fold interquartile range (IQR) and the lower whisker the 25th percentile minus the 1.5-fold IQR. In (a) the average temperature is given in green.

## 4 Conclusions

An extension of the MARGA analysis was described to quantify online low-molecular weight organic acids. Therefore, the MARGA was combined with a new setup consisting of an autosampler and a Compact-IC with internal pre-concentration. Laboratory optimizations of the Compact-IC were performed to improve the separation of the target organic acids formate,

acetate, propionate, butyrate, glycolate, pyruvate, oxalate, malonate, succinate, malate, glutarate, and methanesulfonate. An upgrade to a gradient system and an extension of the Metrosep A Supp 16 column to a total length of 400 mm allowed for a satisfactory separation of all MCAs and DCAs with low limits of detection and precisions.

The example application of the system in May 2017 illustrated high concentrations of formic acid and oxalate in the late afternoon, indicating a photochemical formation by atmospheric precursors. Variations of the wind direction resulted in sudden

changes in the concentrations, as was the case for methanesulfonate.

To the author's knowledge, a high resolved data of low-molecular weight organic acids are not available for rural Central Europe. Before our investigation, a quantification of these acids in the particle-phase was only possible with filter measurements resulting in a low time resolution and potential artifacts from adsorption or revolatilization. The results of the example application proved the suitability of the MARGA extension for field measurements. Compared to other online

systems, the variety of quantifiable organic acids in the gas and particle-phase is unique. The application of this online method reduces laboratory work and sampling artifacts by filter and impactor measurements. Additionally, obtaining every second hour information of the organic acids allowed for the investigation of diurnal cycles, improving the knowledge of their primary and/or secondary sources. For the investigation of tropospheric multiphase chemistry, simultaneous quantification of the gas and the particle-phase concentrations promises interesting analyses of the phase distribution of each organic acid.

**Data availability**

Data can be made available from the authors upon request.

**Author contribution**

HH provided the concept for the MARGA extension. BS performed the experimental development, the calculations, the combination in the field, the measurements and wrote the manuscript. GS, DvP and HH contributed ideas and suggestions

during the method development and the field measurements. AG helped during infrastructural issues in Melpitz. All authors provided additional input and comments during the preparation of the manuscript.

**Competing interests**

The authors declare that they have no conflict of interest.

## Acknowledgements

We thank R. Rabe for the support, especially in the field. The authors acknowledge financial support for this study and the deployment of the MARGA system from the German Federal Environment Agency (UBA) research foundation under contract No 52436 as well as from the European Regional Development fund by the European Union under contract No 100188826. This study is partly supported by ACTRIS-2 (Aerosol, Clouds, and Trace gases Research InfraStructure network) from the European Union's Horizon 2020 research and innovation programme under grant agreement No 654109.

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
