# Peer review of "Development of an online-coupled MARGA upgrade for the two hours interval quantification of low-molecular weight organic acids in the gas and particle-phase"

_Atmospheric Measurement Techniques, 2018_

## Referee Comment (RC1) · Anonymous Referee #1 · 30 Oct 2018

This paper discusses modifications to a MARGA for measurement of gas and particle organic acids, along with the traditional inorganic species. This type of data could be highly useful and so there is significant value in such an instrument. However, overall the paper is of marginal value. It is largely about setting up the ion chromatographic system to separate and measure all the various ions in a reasonable time. This information could have been presented in a technical report. Or alternatively, as there is apparently a paper focused on data analysis and interpretation from this study in progress, an option would have been to include the topics discussed in the coming

paper supplemental section.

In any case, the authors background on methods is sparse and some new references are missing. For example, there are other IC approaches (capillary ICs) that can effectively measure these ions, see Nah et al, cited below. Possible artifacts with this method are not discussed, eg, is the denuder 100% efficient at only removing gases? A major shortcoming is that the ambient data, shown in support of the utility of the instrument, looks poor due to lack of data? Why is this? Did the instrument not work during those periods, or was the species to be measured below the LOD? This needs much more discussion. Example, if most of the species were below LOD for much of the time, why all the work to attempt to measure them with an online system? Can one expect to be able to measure them with this instrument at other locations? That is, does this data really demonstrate the value of this instrument, eg, pg 24 line 11 in Conclusion it states: The results of the example application proved the suitability of the MARGA extension for field measurements. In my view the lack of measurement of most species proves it did not perform well – an instrument that runs but provides little data is likely not of much value. These issues must be addressed prior to publication. Maybe there is another explanation for the lack of ambient data?

Specific Comments.

Introduction: No discussion of recent organic acid aerosol paper, which is directly pertinent. (Nah, T., H. Guo, A. P. Sullivan, Y. Chen, D. J. Tanner, A. Nenes, A. Russell, N. L. Ng, L. G. Huey, and R. J. Weber (2018), Characterization of Aerosol Composition, Aerosol Acidity and Water-soluble Organic Acids at an Agriculture-intensive Rural Southeastern U.S. Site, Atm. Chem. Phys., 18, 11471-11491.)

Pg 2 line 28 and on. CIMS is criticized due to cost, too much data (is that really a problem?) and need for skilled operator. One could argue the same for the MARGA system proposed here. Give numbers for comparison. The MARGA system also likely has the disadvantage that it needs constant attention, unlike a CIMS. This should be

clarified.

Methods: For cases, such as the measurement of organic acids with the described instrument, where a single detection method is used that cannot distinguish between gas or particle phases of a species and the species exists mainly in one phase, less than perfect gas/particle separation can lead to large measurement errors. For example, in this arrangement with the measurements done in series, gas then particle, say most of species X was in the particle phase (i.e., particle » gas), even a very small percent absorption of particles in the denuder will result in large bias in the gas phase measurement. In the other case, particle « gas, less than 100% gas collection will lead to a large bias in particle measurements, if the particle collection system will also collect the gas, which is likely in this system. For the study of organic acids, this is a large issue, which should be discussed in detail, that is report the denuder gas collection efficiency with uncertainty, and the penetration of particles through the denuder.

Results: What is the cause for all the missing data in Fig 8, data below LOD? Is so this should be discussed. Ie, give report fraction of data above LOD for all species in the ambient study.

Page 22, Fig 9. Maybe the diurnal profile was not due to photochemical formation, but instead temperature?

Pg 23, line 19-20. If most of the data is below LOD, how can the authors claim the instrument was a success?

---

## Referee Comment (RC2) · Anonymous Referee #2 · 7 Nov 2018

This paper presents a method for making hourly integrated gas and particle-phase organic acid measurements. This is accomplished by extending a Monitor for AeRosols and Gases in ambient Air (MARGA) system to include an additional Compact ion chromatograph. The details of the organic acid column testing and how the extension to the MARGA works are provided. Example application data from measurements made in Melpitz, Germany are presented.

Overall this is a good paper. I really just have a number of comments to help with the flow of the paper. All of these are outlined below and need to addressed before the

paper can be considered for publication.

General Comments -I think it would be helpful throughout the paper to refer to the Compact-IC as the additional Compact-IC or additional IC-system. This would really help to separate it from the ICs that are part of the original MARGA set-up.

Specific Comments First line of Title – Is two-hourly really correct? My initial thought was that it should be two hour integrated measurements. But actually isn't it that the hourly integrated sample from the MARGA is then additionally measured 1 hour later for organic acid. So then I think it is actually hourly integrated quantification of low-molecular weight organic acids.

Abstract Page 1, Line 12 – Same as above. I think two-hourly time resolution should be one hour integrated measurement

Page 1, Line 15 – Suggest removing the for before gradient

Page 1, Line 25 – Suggest removing the a after indicate

Page 1, Line 26 – Believe that something is missing at the end of the sentence. Should it be as a source or was present?

Introduction Page 1, Line 28 – Suggest changing were measured to have been measured

Page 2, Line 2 – Suggest changing formed secondary to formed as secondary products

Page 2, Line 7 – Suggest changing have a sensitive influence on the ecosystem to have an influence on a sensitive ecosystem

Page 2, Line 21 – (Steiger et al., 2018) should be Steiger et al. (2018)

Page 2, Line 34 – Suggest changing filter to filters

Page 3, Line 2 – Suggest changing resolution of to resolution from

Page 3, Line 3 – Suggest changing filter to filters

Page 3, Line 6 – Suggest changing A detection to The detection

Page 3, Line 11 – Suggest changing Therefore to In this case

Page 3, Line 12 – Suggest changing is limited to was limited

Page 3, Line 18 – Suggest changing applied to employed

Page 3, Line 19 – Suggest changing of successful field application, first to of a successful field application, the first

Page 3, Line 20 – Suggest adding an a before focus

2.Instrumentation and materials Page 3, Line 29 – Should Rotation be Rotating?

Page 3, Line 30 – Suggest adding a the before WRD

2.2.Additional IC system Page 4, Line 6 – Believe the word compounds should be components

Page 4, Line 7 – Suggest changing the as after systems to a comma, adding a comma after Scientific, and putting a period after alternatives

Page 4, Lines 8-9 – Suggest having a new sentence begin with But the liquid. Also suggest adding a comma before especially and after autosampler

Page 4, Line 29 – Suggest adding a the before MARGA

Page 5, Line 8 – Suggest adding the phrase in the standard after ion

3.Results and discussion 3.1.Development of the IC separation Page 7, Line 7 – Suggest adding a the before WRD

Page 7, Line 10 – What is the eluent used for the isocratic separation on the Metrosep A Supp 16 250 mm column? I believe that only the chemicals used have previously been mentioned and not the eluent.

Page 7, Lines 26-27 – The authors mention that it could be expected that the separation would worsen for high concentrations with this anion-exchange column. But no explanation for this is provided. I think it would be helpful to add some text for readers not as familiar with chromatography.

Page 8, Line 11 – Suggest changing tailing to tail

Page 9, Line 9 – Suggest changing differently to different

Page 9, Lines 10-11 – The authors list the Na2CO3 part of the eluent, but not the NaOH part. I think it would be more accurate to include it since that both eluents A and B have to be made as a mixture.

Page 9, Line 13 – Suggest changing analysis tome before of the F- peak to time before the F- peak eluted. Also suggest removing the the before eluent

Page 9, Line 15 – Suggest changing succeeding to successful

Page 10, Table 2, first line of caption – Suggest adding a the before varied, changing column to the, and adding the words column along after 250 mm

Page 11, Figure 4, third line of caption – Suggest adding a the before green

Page 11, Line 10 – Suggest changing tailing to tail

Page 11, Line 11 – Suggest changing Change to Changing

Page 12, Line 1 - Suggest changing prolonged to extended

Page 12, Line 3 – Suggest changing Resulting from the to Due to the. Also suggest removing the word long before coupled

Page 12, Line 5 - Suggest changing prolonged to extended

Page 12, Line 8 – Suggest changing carryovers to carryover and adding a for before starting

Page 12, Line 11 – Suggest removing the the before eluent

Page 12, Line 13 – Suggest changing solutions, but was to solutions and was

Page 12, Lines 13-14 - Suggest changing by the used chemicals or the used glassware is likely to from the chemicals or glassware used is likely

Page 12, Figure 5, first line of caption – Suggest adding the word columns after 150 mm and elution after gradient

Page 12, Figure 5, third line of caption – Suggest removing the phrase with a gradient system

Page 12, Figure 5, fourth line of caption – Suggest adding a the before green

3.2.Limits of detection and precision Page 14, Line 11 – (Funk et al., 2005) should be Funk et al. (2005)

Page 15, Line 2 – Suggest adding a the before case and removing the is before F

Page 15, Line 9 – Suggest adding a the before quotient

Page 15, Line 10 – Suggest removing the phrase results in the standard deviation of the method as it is stated previously in the sentence

Page 15, Line 12 – A colon is missing after given by

Page 16, Line 10 – A colon is missing after x = 0 is

Page 16, Line 12 – A colon is missing after xLOD

Page 17, Line 1 – A colon is missing after T is

Page 17, Line 9 – Suggest changing of peak areas to of the peak area

3.3.Sample handling Page 18, Line 5 – Suggest changing display to displays

Page 18, Line 12 – Suggest changing solutions were directed into the waste to solution was directed to waste

Page 19, Figure 7 - Might suggest changing either the black or blue line in plot b to another color as these two look very similar

Page 19, Figure 7, Fourth line of caption – Suggest adding a the before new

Page 20, Line 1 – Believe a two-hourly time resolution should be a hour integrated measurement

Page 20, Line 3 – Suggest adding the word column after pre-concentration

3.4.MARGA absorption solution Page 20, Line 13 – Suggest adding a the before absorption

Page 20, Line 14 – Suggest changing Metrohm-Applikon, the Netherlands, allowed integrating the to Metrohm-Applikon (Netherlands) allowed for integration of the

Page 20, Line 19 – The authors mention that the absorption solution in the MARGA was replaced with a 1% H2O2 solution. But what is the typical solution used? It is not actually mentioned and this would be helpful to note since the authors are saying that it is important that the concentration be changed to add the additional analysis of organic acids.

3.5.Intercomparison of inorganic ions Page 20, Line 23 – Suggest adding a the before MARGA

Page 21, Line 1 – It should be a R2

Page 21, Table 4, First line of caption – Suggest adding a the before MARGA

Page 21, Line 19 – Suggest adding a the before MARGA

3.6.Example application in the field Page 23, Line 8 – Suggest changing averaged concentration in this to average concentration over this

Page 23, Line 9 – Suggest changing averaged to average

Page 23, Line 20 – Suggest adding a the before gas

Page 23, Line 21 – There should be no hyphen between one and year

4.Conclusions Page 24, Line 10 – Suggest removing the it before was the case

Page 24, Lines 13-14 – the time resolution is listed as two hours, but I believe it is actually hour integrated

Page 24, Line 14 – Suggest adding a for before the investigation

Data availability Page 24, Line 18 – Suggest changing from authors on request to from the authors upon request

Author contribution Page 24, Line 20 – Suggest changing concepted to provided the concept for

Acknowledgements Page 25, Line 2 – Suggest changing support of to support for

Page 25, Line 3 – Suggest adding a the before deployment and changing system by to system from

References Page 26, Line 6 – Believe Rondonia should have an accent mark

Page 27, Line 20 – Believe Gelencser is missing accent marks

Supporting Information Page 15, Figure S15 - The same gradient program as for all the other tests is being used, correct? If so, then I might suggest just saying that in the caption. But if the authors do want to keep the program in the corner of the plot then it should probably say %B so it more clear.

Page 15, Figure S15 - Pyruvate/bromide and oxalate are mentioned in the text, but they are not actually labeled in the figure. It might be helpful to include them for the reader.

Page 18, Figure S17, first line of caption – Suggest adding a the before MARGA

Page 18, Figure S17, second line of caption – Suggest changing during one-year measurement to during the one year long measurement

Page 19, Figure S18, first line of caption – Suggest adding a the before MARGA

Page 19, Figure S18, second line of caption – Suggest changing one-year measurement to one year long measurement

Page 10, Figure S19 – RH and P on the right hand y-axis should be capitalized

---

## Author Comment (AC1) · 20 Dec 2018

**Comments on RC1:**

First, we thank the referee for the careful reading and the helpful comments on our manuscript. We pasted the referee comments below (**bold**), followed by our author responses in-line and refer to our manuscript in the uploaded version of 10[th] October 2018.

**This paper discusses modifications to a MARGA for measurement of gas and particle organic acids, along with the traditional inorganic species. This type of data could be highly useful and so there is significant value in such an instrument. However, overall the paper is of marginal value. It is largely about setting up the ion chromatographic system to separate and measure all the various ions in a reasonable time. This information could have been presented in a technical report. Or alternatively, as there is apparently a paper focused on data analysis and interpretation from this study in progress, an option would have been to include the topics discussed in the coming paper supplemental section.**

We thank the reviewer for generally agreeing on the value of our developed method. We have considered the suggestion to combine method development and data analysis in one large paper. However, we consider the modification of the commercial MARGA system and especially the development and validation of the chromatographic separation to be important enough to justify their stand-alone publication, together with an example application. The ion chromatographic separation of the reported acids with sufficient resolution from potentially interfering organic and inorganic ions has actually been quite an analytical challenge and we think it is beneficial to readers to report these developments in detail together with the final method that promises the reliable quantification of the dominant mono- and dicarboxylic acids as well as methanesulfonic acid in the gas and particle-phase. We also think that the described work is fully in line with the scope of AMT, i.e. "the development, intercomparison, and validation of measurement instruments", and would therefore like to publish it as a full paper.

**In any case, the authors background on methods is sparse and some new references are missing. For example, there are other IC approaches (capillary ICs) that can effectively measure these ions, see Nah et al, cited below.**

We apologize for this oversight and have now included the mentioned paper in the discussion as will be shown in the specific comments below.

**Possible artifacts with this method are not discussed, eg, is the denuder 100% efficient at only removing gases?**

We agree this is an important point in the evaluation of this system especially regarding the gas and particle-phase contributions in the later interpretation. We inserted a complete new chapter dealing about this topic. Please find more information in the specific comments below.

**A major shortcoming is that the ambient data, shown in support of the utility of the instrument, looks poor due to lack of data? Why is this? Did the instrument not work during those periods, or was the species to be measured below the LOD? This needs much more discussion.**

In Figure 8, we have now indicated the few periods when the Compact-IC was not working due to cleaning, blank measurements and calibration. Missing concentrations were cut because of high LOD determined after a rather conservative and theoretical German VDI norm. For a better comparison with other instruments, we changed the approach for the LOD estimation. Please

find more information in the specific comments. Additionally, we now discuss possible reasons for the low concentrations in the completely revised chapter 3.7 about the example application in the field.

**Example, if most of the species were below LOD for much of the time, why all the work to attempt to measure them with an online system?**

The large fraction of below LOD data has also in part resulted from a rather conservative estimation of the LOD according to the German VDI norm. As we now explain in the revised version of section 3.7, the data coverage during the example application period significantly increases if we estimate LODs based on the internationally more established 3σ criterion.

**Can one expect to be able to measure them with this instrument at other locations?**

More complete data coverage can of course be expected for locations with higher concentrations of the target acids. Also for the Melpitz location, it can be expected that during periods with high precursor concentrations and stronger photochemical conditions or during periods with stronger anthropogenic influence the concentrations will be higher and the MARGA upgrade will be able to measure them.

**That is, does this data really demonstrate the value of this instrument, eg, pg 24 line 11 in Conclusion it states: The results of the example application proved the suitability of the MARGA extension for field measurements. In my view the lack of measurement of most species proves it did not perform well – an instrument that runs but provides little data is likely not of much value.**

We understand these doubts but would like to argue that even somewhat incomplete data is better than no data at all. Our aim was to investigate the seasonal cycle of the organic acids in the gas and particle-phase and we developed an instrument that is able to analyze these compounds with a good chromatographic separation. We think this aim has been met even if not all of the organic acids could always be detected above LOD at the given site. A longer-term study of gaseous and particle-bound organic acid concentrations in rural Central Europe with high time resolution has not existed so far and even the sometimes low concentrations of the organic acids can help to classify the Melpitz site in the comparison with other locations around the world.

**These issues must be addressed prior to publication. Maybe there is another explanation for the lack of ambient data?**

We agree these issues were not addressed to a sufficient extent in the original submission and therefore discuss them in more detail in the revised version, as can also be seen in the specific comments below.

**Specific Comments.**

**Introduction: No discussion of recent organic acid aerosol paper, which is directly pertinent. (Nah, T., H. Guo, A. P. Sullivan, Y. Chen, D. J. Tanner, A. Nenes, A. Russell, N. L. Ng, L. G. Huey, and R. J. Weber (2018), Characterization of Aerosol Composition, Aerosol Acidity and Water-soluble Organic Acids at an Agriculture-intensive Rural Southeastern U.S. Site, Atm. Chem. Phys., 18, 11471-11491.)**

Thank you for giving references to this interesting paper. We compared our field measurements with this paper and added the following paragraph on Page 3, Line 14:

"Recently, Nah et al. (2018a) presented measurements of low-molecular weight organic acids within the gas and particle-phase with use of a CIMS and a Particle-Into-Liquid-Sampler (PILS) coupled with a capillary high-pressure ion chromatography (HPIC), respectively. They reported hourly concentrations of these compounds in a rural southeastern United States site for two months and were able to investigate the gas-particle partitioning."

To further complete the literature study, we also add Ullah et al. (2006) below.

"Ullah et al. (2006) developed an on-line instrument to measure ionic species within the gas and particle-phase. For the separation, they used a membrane denuder to collect the water-soluble gases and a hydrophilic filter sampled the particles. In their ion chromatography analysis, it was possible to quantify formic and acetic acid every 40 minutes.
However, to the author's knowledge,…"

We inserted Hu et al. (2018) as representative for GC-MS analysis on page 2, line 12.
We inserted Mungall et al. (2018) as representative for CIMS measurements on page 2, line 26.
We inserted Deshmukh et al. (2018) as representative fro GC-FID on page 2, line 13:

"…or flame ionization detector (GC-FID) (Deshmukh et al., 2018),…"

Additionally, we included all new references in the reference list.

**Pg 2 line 28 and on. CIMS is criticized due to cost, too much data (is that really a problem?) and need for skilled operator. One could argue the same for the MARGA system proposed here. Give numbers for comparison. The MARGA system also likely has the disadvantage that it needs constant attention, unlike a CIMS. This should be clarified.**

The reviewer is completely right here. Boring et al. (2002) listed these requirements of MS based measurements, but we agree that the MARGA does not have significant advantages over these in terms of costs or operator skills. We therefore deleted this paragraph on page 2, lines 29-31.

The MARGA system does have some advantages in terms of integrated and parallel gas- and particle phase sampling, however. In addition, it is well suited for longer-term measurements, as it is typically applied as a stationary instrument. Also, chromatographic separation and quantification results in highly confident data, as we can practically rule out significant interferences from other species. This can be an issue with online MS instruments, where species with the same or similar masses can bias the quantification of the target species.

Certainly, MS-based and more traditional liquid-based online instruments both have their justification in certain application areas. As we would like to avoid a deep comparative discussion of their strengths and limitations in this manuscript we now solely focus on the benefits of the MARGA system in the revised paper.

Accordingly, we rewrote on page 3, line 16:

"The present study describes the instrumental development of an online-coupled pre-concentration and ion chromatographic (IC) separation system to determine organic acids in the gas and particle-phase as an extension of the MARGA. The MARGA has been reported a reliable field instrument for long-time measurements in Melpitz and other sites (Stieger et al., 2018 and references therein) and its upgrade with an additional IC separation allows for the analysis of all target compounds with low risk of interferences from other species."

**Methods: For cases, such as the measurement of organic acids with the described instrument, where a single detection method is used that cannot distinguish between gas or particle phases of a species and the species exists mainly in one phase, less than perfect gas/particle separation can lead to large measurement errors. For example, in this arrangement with the measurements done in series, gas then particle, say most of species X was in the particle phase (i.e., particle » gas), even a very small percent absorption of particles in the denuder will result in large bias in the gas phase measurement. In the other case, particle « gas, less than 100% gas collection will lead to a large bias in particle measurements, if the particle collection system will also collect the gas, which is likely in this system. For the study of organic acids, this is a large issue, which should be discussed in detail, that is report the denuder gas collection efficiency with uncertainty, and the penetration of particles through the denuder.**

This is a good point and we agree it should be discussed in the manuscript to evaluate this system. We used theoretical approaches to calculate the annular denuder efficiencies of each organic acid. High collection efficiencies of over 99% were mostly determined. When applying the resulting denuder efficiencies and the highest gas phase concentrations of each organic acids, the worst-case artifact particulate concentrations are close to the LOD and do therefore not strongly bias the comparisons between gas and particle-phase. For more detailed information we inserted a new chapter "3.5 WRD efficiency and particle penetration":

**"3.5 WRD efficiency and WRD particle collection**

[revised manuscript text omitted]

We added additional information in the Supplement:

**WRD efficiency**

$d_i$   -   inner diameter (4.2 cm)
$d_o$   -   outer diameter (4.5 cm)
$d$   -   hydrodynamic equivalent diameter ($d_o - d_i = 0.3$ cm)
$L$   -   length of the denuder (30 cm)
$D$   -   diffusion coefficient (calculated according to Fuller et al. (1966))
$u$   -   flow velocity (16.7 l min$^{-1}$)
$E$   -   denuder efficiency

**Table S2.** Equations for the calculations of the efficiencies (E) for annular denuders.

| | Winiwarter (1989) | Possanzini et al. (1983) | De Santis (1994) | Berg et al. (2010) |
|---|---|---|---|---|
| X | $\dfrac{2LD}{d^2 u}$ | $\dfrac{\pi LD(d_i + d_o)}{4ud}$ | $\dfrac{\pi LD(d_i + d_o)}{ud}$ | Efficiencies were calculated with their described spreadsheet calculator |
| E | $1 - 9.11 \cdot e^{-3.884^2 X}$ | $1 - 0.82 \cdot e^{-22.53X}$ | $1 - 0.91 \cdot e^{-7.54X}$ | |

We included new citations in the reference list.

Due to the new Table 4, we changed the old Table 4 to Table 5 on:
Page 20, Line 27,
Page 21, Line 12

Due to the new chapter, we changed chapter 3.5 to 3.6 and 3.6 to 3.7

**Results: What is the cause for all the missing data in Fig 8, data below LOD? Is so this should be discussed. Ie, give report fraction of data above LOD for all species in the ambient study.**

The missing concentrations were predominantly below the LOD. However, we realised that the originally applied German VDI norm represents a rather conservative approach for the calculation of the LOD and resulted in rather high LODs. When including all below LOD values that were originally removed from the data set, we observed much less missing data and much better data coverages. We have therefore decided to replace the originally applied VDI LOD calculation approach with an estimation approach that is internationally more established. We now calculate the LOD from mean blank values plus three times the standard deviation ($3\sigma$) or, for acids with no blank signals, from the smallest observable peaks. These LODs are lower than the originally reported ones and range between 0.5 ng m$^{-3}$ for malonate and 17.4 ng m$^{-3}$ for glutarate.

Although the data coverage was improved by applying the $3\sigma$ method, some values for several organic acids were still below the LOD or were not detected. As additional information, we included in Figure 8 periods in grey, in which no data were available because of instrumental issues like cleaning procedures, blank measurements or calibration of the Compact-IC.
Possible reasons for not detectable concentrations are changeable weather conditions, lower temperatures, and cloudy weather during the first days, leading to lower emissions of the acids or to less effective formation from their atmospheric precursor species.

We completely revised the chapter 3.7 Example application in the field. In Figure 8, we marked periods with grey where no data are available because of instrumental issues. We added Table 6 with the percental data coverages of each organic acid during the example measurement period. In the newly revised chapter, we explain the low concentrations as follows:

[revised manuscript text omitted]

New references were included in the reference list.

**Page 22, Fig 9. Maybe the diurnal profile was not due to photochemical formation, but instead temperature?**

You are right. The direct comparison of the average temperature and the diurnal behaviour of formic acid and oxalate show very good agreements. During night, the surface cools down. Consequently, the air temperature above the surface decreases leading to a near-ground inversion. In this case, a vertical mixing is prohibited and the concentrations decreases due to deposition. The daily course is discussed within the revised chapter 3.7. As graphical overview, we inserted the average temperature in Figure 9a.

[Figure]

**Figure 9.** Box-Whisker-Plot for the diurnal variation of gaseous formic acid (a) and particulate oxalate (b) in Melpitz for the example application in the field. The red dots represent the mean, the box the 25th and 75th percentile and the upper whisker the 75th percentile plus the 1.5-fold interquartile range (IQR) and the lower whisker the 25th percentile minus the 1.5-fold IQR. In (a) the average temperature is given in green.

**Pg 23, line 19-20. If most of the data is below LOD, how can the authors claim the instrument was a success?**

We understand this criticism. We changed our LOD calculation to the internationally better known $3\sigma$ approach also to make the described setup more comparable to others. The new approach resulted in an improved data coverage of the organic acids (up to 100%). Not detectable concentrations for some DCAs in the particle-phase and some MCAs in the gas-phase are a result of low concentrations during the studied period due to lower temperatures and cloudy weather.

However, also very low concentrations during the example application are part of the result of a one-year measurement campaign to investigate the seasonal course of different organic acids in Melpitz. Additionally, these low concentrations help us to evaluate Melpitz in comparison with stations around the world.

The described method showed a good intercomparison between MARGA and Compact-IC indicating a trustworthy calibration. The separation of the gas and particle-phase by the WRD/SJAC setup within the MARGA show negligible interferences.

All the findings in the revised manuscript let us claim that the instrument was a success.

Additionally we added on page 24, line 11:

"To the author's knowledge, high resolved data of low-molecular weight organic acids are not available for rural Central Europe. Before our investigation, a quantification of these acids in the particle-phase was only possible with filter measurements resulting in a low time resolution and potential artifacts from adsorption or revolatilisation."

---

## Author Comment (AC2) · 20 Dec 2018

**Comments on RC2:**

First, we thank the referee for the careful reading and the helpful comments on our manuscript. We pasted the referee comments below (**bold**), followed by our author responses in-line and refer to our manuscript in the uploaded version of 10th October 2018.

**This paper presents a method for making hourly integrated gas and particle-phase organic acid measurements. This is accomplished by extending a Monitor for AeRosols and Gases in ambient Air (MARGA) system to include an additional Compact ion chromatograph. The details of the organic acid column testing and how the extension to the MARGA works are provided. Example application data from measurements made in Melpitz, Germany are presented.**

**Overall this is a good paper. I really just have a number of comments to help with the flow of the paper. All of these are outlined below and need to addressed before the paper can be considered for publication.**

We thank the reviewer for his positive overall judgement.

**General Comments**

**General Comments - I think it would be helpful throughout the paper to refer to the Compact-IC as the additional Compact-IC or additional IC-system. This would really help to separate it from the ICs that are part of the original MARGA set-up.**

This is a good point. We will refer to Compact-IC throughout the manuscript. We changed:

Page 1, Line 13
Page 1, Line 20
Page 4, Line 4
Page 18, Line 3
Page 19, Line 3 (Figure caption)
Page 19, Line 6 (Figure caption)

**Specific Comments First line of Title – Is two-hourly really correct? My initial thought was that it should be two hour integrated measurements. But actually isn't it that the hourly integrated sample from the MARGA is then additionally measured 1 hour later for organic acid. So then I think it is actually hourly integrated quantification of low-molecular weight organic acids.**

This is right. We measured the organic acids from hourly integrated MARGA samples but we need two hours for the analysis of the gas and particle-phase. That is why we collect one hourly integrated sample every two hours. Every other hour, the MARGA sample outflow is discarded. Thus, we achieve 12 samples per day for WRD and SJAC each. We added this information also later in the Sample handling chapter. We changed the title and hope it is less confusing:

"Development of an online-coupled MARGA upgrade for the two hours interval quantification of low-molecular weight organic acids in the gas and particle-phase"

**Abstract Page 1, Line 12 – Same as above. I think two-hourly time resolution should be one hour integrated measurement**

We add a sentence on page 1, line 13:

"Therefore, every second hourly integrated MARGA gas and particle-sample were collected and analyzed by the Compact-IC resulting in 12 values per day for each phase."

**Page 1, Line 15 – Suggest removing the for before gradient**

We deleted the "for".

**Page 1, Line 25 – Suggest removing the a after indicate**

We deleted the "a".

**Page 1, Line 26 – Believe that something is missing at the end of the sentence. Should it be as a source or was present?**

You are right, something was missing. We added at the end of the sentence "…formation as a source."

**Introduction Page 1, Line 28 – Suggest changing were measured to have been measured**

We replaced "were measured" with "have been measured".

**Page 2, Line 2 – Suggest changing formed secondary to formed as secondary products**

We rewrote this to "…formed as secondary products…".

**Page 2, Line 7 – Suggest changing have a sensitive influence on the ecosystem to have an influence on a sensitive ecosystem**

We rewrote the sentence to: "…can have an influence on the sensitive ecosystem…"

**Page 2, Line 21 – (Stieger et al., 2018) should be Stieger et al. (2018)**

(Stieger et al., 2018) is correct but we can imagine that this is irritating. We rewrote the sentence:

"Recently, Stieger et al. (2018) showed that off-line filter analysis involves the risk of possible evaporation artifacts of volatile particulate compounds from the filter or the adsorption of gaseous compounds. Additionally, Boring et al. (2002) mentioned the difficulty of sampling very small particles by impaction techniques."

**Page 2, Line 34 – Suggest changing filter to filters**

We changed it to "filters".

**Page 3, Line 2 – Suggest changing resolution of to resolution from**

We replaced "of" by "from".

**Page 3, Line 3 – Suggest changing filter to filters**

We changed "filter" to "filters".

**Page 3, Line 6 – Suggest changing A detection to The detection**

We changed "A detection" to "The detection".

**Page 3, Line 11 – Suggest changing Therefore to In this case**

We changed "Therefore" to "In this case".

**Page 3, Line 12 – Suggest changing is limited to was limited**

We changed "is limited" to "was limited".

**Page 3, Line 18 – Suggest changing applied to employed**

We changed "applied" to "employed".

**Page 3, Line 19 – Suggest changing of successful field application, first to of a successful field application, the first**

We rewrote the phrase to "As a demonstration of a successful field application, the first…"

**Page 3, Line 20 – Suggest adding an a before focus**

We added an "a".

**2.Instrumentation and materials Page 3, Line 29 – Should Rotation be Rotating?**

You are right. We changed it to "Rotating".

**Page 3, Line 30 – Suggest adding a the before WRD**

We added "the".

**2.2.Additional IC system Page 4, Line 6 – Believe the word compounds should be components**

We changed "compounds" to "components".

**Page 4, Line 7 – Suggest changing the as after systems to a comma, adding a comma after Scientific, and putting a period after alternatives**
**Page 4, Lines 8-9 – Suggest having a new sentence begin with But the liquid. Also suggest adding a comma before especially and after autosampler**

We rearranged the sentences to "Comparable IC systems, for example from Thermo Scientific, were considered as possible alternatives. However, the liquid handling via the autosampler, especially the liquid flows from the MARGA to the necessary autosampler and the capacity of the autosampler, limited the use of other IC systems."

**Page 4, Line 29 – Suggest adding a the before MARGA**

We added a "the".

**Page 5, Line 8 – Suggest adding the phrase in the standard after ion**

We added the suggested phrase.

**3.Results and discussion 3.1.Development of the IC separation Page 7, Line 7 – Suggest adding a the before WRD**

We added a "the".

**Page 7, Line 10 – What is the eluent used for the isocratic separation on the Metrosep A Supp 16 250 mm column? I believe that only the chemicals used have previously been mentioned and not the eluent.**

We included the eluent concentration and rearranged the sentences to: "First analyses were performed with an isocratic system and the separation column Metrosep A Supp 16 250 mm with an eluent of 7 mM $Na_2CO_3$ and 0.75 mM NaOH. The resulting chromatogram is shown in Fig. 2,…"

**Page 7, Lines 26-27 – The authors mention that it could be expected that the separation would worsen for high concentrations with this anion-exchange column. But no explanation for this is provided. I think it would be helpful to add some text for readers not as familiar with chromatography.**

We added a further sentence.

"Regarding the low standard concentrations, the separation can be expected to worsen for high concentrations with this anion-exchange column. Higher ion concentrations would broaden the single peaks, which leads to co-elution."

**Page 8, Line 11 – Suggest changing tailing to tail**

We replaced "tailing" by "tail".

**Page 9, Line 9 – Suggest changing differently to different**

We replaced "differently" by "different".

**Page 9, Lines 10-11 – The authors list the Na2CO3 part of the eluent, but not the NaOH part. I think it would be more accurate to include it since that both eluents A and B have to be made as a mixture.**

You are right. We added the information.

**Page 9, Line 13 – Suggest changing analysis tome before of the F- peak to time before the F- peak eluted. Also suggest removing the the before eluent**

We rewrote the sentence.

"In this example, the fraction of eluent B was increased to 50% shortly before the beginning of the analysis to shorten the analysis time before the $F^-$ peak eluted. At retention time t = 5 min eluent B was set to 0 %,…"

**Page 9, Line 15 – Suggest changing succeeding to successful**

We wanted to express that the decrease to 0% is necessary for the next (following) analysis. We hope that the replacement of "succeeding" with "subsequent" is more clear.

**Page 10, Table 2, first line of caption – Suggest adding a the before varied, changing column to the, and adding the words column along after 250 mm**

We changed everything.

"Overview of the varied flows and eluent compositions in the isocratic system using the Metrosep A Supp 16 250 mm column with their effects on separation and reference to the corresponding figures in the supplement."

**Page 11, Figure 4, third line of caption – Suggest adding a the before green**

We added "the".

**Page 11, Line 10 – Suggest changing tailing to tail**

We changed "tailing" to "tail".

**Page 11, Line 11 – Suggest changing Change to Changing**

We corrected "Change" to "Changing".

**Page 12, Line 1 - Suggest changing prolonged to extended**

We changed "prolonged" to "extended".

**Page 12, Line 3 – Suggest changing Resulting from the to Due to the. Also suggest removing the word long before coupled**

We rewrote the sentence.

"Due to the increased back-pressure of the coupled columns,…"

**Page 12, Line 5 - Suggest changing prolonged to extended**

We changed "prolonged" to "extended".

**Page 12, Line 8 – Suggest changing carryovers to carryover and adding a for before starting**

We corrected this to: "…led to a carryover of…" and "…was found for starting…"

**Page 12, Line 11 – Suggest removing the the before eluent**

We deleted the "the".

**Page 12, Line 13 – Suggest changing solutions, but was to solutions and was**

We replaced "…, but…" with "…and…"

**Page 12, Lines 13-14 - Suggest changing by the used chemicals or the used glassware is likely to from the chemicals or glassware used is likely**

We rearranged the sentence to: "Thus, a contamination from the chemicals or glassware used is likely."

**Page 12, Figure 5, first line of caption – Suggest adding the word columns after 150 mm and elution after gradient**

We inserted both words.

**Page 12, Figure 5, third line of caption – Suggest removing the phrase with a gradient system**

We deleted this phrase.

**Page 12, Figure 5, fourth line of caption – Suggest adding a the before green**

We added "the".

**3.2.Limits of detection and precision Page 14, Line 11 – (Funk et al., 2005) should be Funk et al. (2005)**

Again, this is a little bit irritating in the manuscript. We rewrote this phrase to: "…was calculated after Funk et al. (2015):"

**Page 15, Line 2 – Suggest adding a the before case and removing the is before F**

We added a "the" and removed the "is".

**Page 15, Line 9 – Suggest adding a the before quotient**

We removed this part from the manuscript. As we used the $3\sigma$ approach for the determination of the LOD.

**Page 15, Line 10 – Suggest removing the phrase results in the standard deviation of the method as it is stated previously in the sentence**

We removed this part from the manuscript.

**Page 15, Line 12 – A colon is missing after given by**

We removed this part from the manuscript.

**Page 16, Line 10 – A colon is missing after x = 0 is**

We removed this part from the manuscript.

**Page 16, Line 12 – A colon is missing after xLOD**

We removed this part from the manuscript.

**Page 17, Line 1 – A colon is missing after T is**

We removed this part from the manuscript.

**Page 17, Line 9 – Suggest changing of peak areas to of the peak area**

We corrected this to "…of the peak area…".

**3.3.Sample handling Page 18, Line 5 – Suggest changing display to displays**

We added the "s".

**Page 18, Line 12 – Suggest changing solutions were directed into the waste to solution was directed to waste**

As two solutions (WRD and SJAC) were directed to waste, we keep the plural but changed "into the waste" to "to waste". Additionally we clarified that both solutions are meant and include this information in the sentence as follows: "…transferred the samples from the WRD and SJAC to the…". Please see also the authors comment below when add additional information to the chapter.

**Page 19, Figure 7 - Might suggest changing either the black or blue line in plot b to another color as these two look very similar**

We changed the color from black to orange.

**Page 19, Figure 7, Fourth line of caption – Suggest adding a the before new**

We added a "the".

**Page 20, Line 1 – Believe a two-hourly time resolution should be a hour integrated measurement**

We add additional information to the chapter on page 18, line 7 that should improve the understanding:

"…This sampling required one hour and yields 25 ml of sample solution in each of the two syringes.

In the second hour, the MARGA syringe pumps transported the solutions to the IC system within the MARGA to analyze the inorganic compounds in the gas and particle-phase, as well as to the autosampler of the Compact-IC. Thereby, the WRD solution was injected with a flow of 0.417 ml min$^{-1}$ into the MARGA-IC to rinse the sampling lines and to fill the injection loop for the first 13 minutes. Afterwards the analysis of this sample followed for 17 minutes. In the second 30-minutes interval, the SJAC sample was injected and analyzed. Only during the injections into the MARGA-IC of both the WRD and SJAC samples, no solutions were transported via an external 6-way-valve (Fig. 1 (g)) either to the autosampler or to the waste. As the vials in the autosampler had a volume of 12.5 ml, the 6-way-valve transferred the samples from the WRD and SJAC to the autosampler only for the first 45 minutes and the rest of the solutions were directed to waste. In the third hour, the WRD sample was pre-concentrated and was analyzed by the Compact IC. Afterwards, the SJAC sample was pre-concentrated and was analyzed in the fourth hour.

To achieve a pre-concentration and analysis of one sample in one hour, the transfer of analytes from the autosampler to the Compact-IC and the pre-concentration of the sample had to be performed within the remaining 7.5 minutes, as the final Compact-IC analysis described previously needed 52.5 minutes. Therefore, the sample flows were increased to 4 ml min$^{-1}$, which is the maximum for what is allowed for the pre-concentration column. For the quantification of the organic acids with the Compact-IC, the hourly integrated MARGA samples were collected every two hours. …"

**Page 20, Line 3 – Suggest adding the word column after pre-concentration**

We do not want to express the column in this sentence but the process of the pre-concentration. Instead, we added "of the sample" after "pre-concentration".

**3.4.MARGA absorption solution Page 20, Line 13 – Suggest adding a the before absorption**

We added "the".

**Page 20, Line 14 – Suggest changing Metrohm-Applikon, the Netherlands, allowed integrating the to Metrohm-Applikon (Netherlands) allowed for integration of the**

We rewrote the sentence as suggested.

**Page 20, Line 19 – The authors mention that the absorption solution in the MARGA was replaced with a 1% H2O2 solution. But what is the typical solution used? It is not actually mentioned and this would be helpful to note since the authors are saying that it is important that the concentration be changed to add the additional analysis of organic acids.**

We added the original concentration of $H_2O_2$ in the first line of the current chapter and rewrote the sentence: "The original MARGA absorption solution in the denuder and SJAC contains 10 mg $l^{-1}$ $H_2O_2$ to avoid…". Five lines later, we gave the information that we excluded $H_2O_2$ because of potential artifact formation with the organic acids and used ultrapure water as the absorption solution.

**3.5. Intercomparison of inorganic ions Page 20, Line 23 – Suggest adding a the before MARGA**

We added "the".

**Page 21, Line 1 – It should be a R2**

We replaced "an" by "a".

**Page 21, Table 4, First line of caption – Suggest adding a the before MARGA**

We added "the".

**Page 21, Line 19 – Suggest adding a the before MARGA**

We added "the".

**3.6. Example application in the field Page 23, Line 8 – Suggest changing averaged concentration in this to average concentration over this**

We completely revised this chapter.

**Page 23, Line 9 – Suggest changing averaged to average**

We completely revised this chapter.

**Page 23, Line 20 – Suggest adding a the before gas**

We completely revised this chapter.

**Page 23, Line 21 – There should be no hyphen between one and year**

We completely revised this chapter but we removed the hyphen also on page 3, line 20 and on page 20, line 26 to make it consistently.

**4. Conclusions Page 24, Line 10 – Suggest removing the it before was the case**

We removed "it".

**Page 24, Lines 13-14 – the time resolution is listed as two hours, but I believe it is actually hour integrated**

Yes, we collect hourly integrated MARGA samples but only every second hour. This is the reason why we got only every second hour the information and we called this a time resolution of two hours. We additional rewrote the sentence:

"Additionally, obtaining every second hour information of the organic acids allowed for the investigation of diurnal cycles, improving the knowledge of their primary and/or secondary sources."

**Page 24, Line 14 – Suggest adding a for before the investigation**

We included "for".

**Data availability Page 24, Line 18 – Suggest changing from authors on request to from the authors upon request**

We changed the sentence as suggested.

**Author contribution Page 24, Line 20 – Suggest changing concepted to provided the concept for**

We replaced "concepted" by "provided the concept for".

**Acknowledgements Page 25, Line 2 – Suggest changing support of to support for**

We replaced "of" by "for".

**Page 25, Line 3 – Suggest adding a the before deployment and changing system by to system from**

We added a "the" and replaced the "by" by "from".

**References Page 26, Line 6 – Believe Rondonia should have an accent mark**
**Page 27, Line 20 – Believe Gelencser is missing accent marks**

We updated both citations.

**Supporting Information Page 15, Figure S15 - The same gradient program as for all the other tests is being used, correct? If so, then I might suggest just saying that in the caption. But if the authors do want to keep the program in the corner of the plot then it should probably say %B so it more clear.**

The conditions (eluent concentration and flow) of these chromatograms in the supporting information differ from the method described in the main manuscript. This is the reason we plot also the time program in the figure. Additionally, we rewrote the figure caption.

"**Figure S15.** Temperature variation of the column oven for 55 °C (black) and 65 °C (red). Eluent A concentration is 1 mM $Na_2CO_3$ / 0.75 mM NaOH and eluent B is 14 mM $Na_2CO_3$ / 0.75 mM NaOH. Chromatogram of a standard solution with aqueous concentrations of 50 µg l$^{-1}$ for Cl$^-$, NO$_3^-$, SO$_4^{2-}$, 25 µg l$^{-1}$ for NO$_2^-$ and 3 µg l$^{-1}$ for F$^-$, Br$^-$ as well as all organic acids. Eluent flow of 1.0 ml min$^{-1}$."

We added "%B" in the figure time program.

**Page 15, Figure S15 - Pyruvate/bromide and oxalate are mentioned in the text, but they are not actually labeled in the figure. It might be helpful to include them for the reader.**

We added the mentioned compounds in the figure.

**Page 18, Figure S17, first line of caption – Suggest adding a the before MARGA**

We added the "the".

**Page 18, Figure S17, second line of caption – Suggest changing during one-year measurement to during the one year long measurement**

We changed the phrase as suggested.

**Page 19, Figure S18, first line of caption – Suggest adding a the before MARGA**

We added "the".

**Page 19, Figure S18, second line of caption – Suggest changing one-year measurement to one year long measurement**

We changed the phrase as suggested.

**Page 10, Figure S19 – RH and P on the right hand y-axis should be capitalized**

We capitalized "RH" and "P" in Figure S19.

**Additional changes:**

**Page 4, Line 10**

We changed "11 ml" to "12.5 ml"

**Page 10, Line 4-5**

We shifted these two lines above the Table 2.

**Page 25, Line 13**

We updated this reference of Boreddy et al. (2017).

**Page 28, Line 26**

We updated this reference of Röhrl and Lammel (2002).